# EUCLID: LESSONS FOR GEOMETRIC LOW-LEVEL VISUAL PERCEPTION IN MULTIMODAL LLMS

## ABSTRACT

Multimodal large language models (MLLMs) have advanced rapidly in recent years, yet they continue to struggle with *low-level visual perception*—particularly, in accurately identifying and describing geometric relationships within images. In this paper, we first diagnose this shortcoming by introducing a dedicated benchmark, *Geoperception*, which focuses exclusively on evaluating geometric low-level perceptual capabilities that serve as essential prerequisites for higher-level visual reasoning. We then present a comprehensive empirical study that investigates strategies for improving model performance in this setting, making use of synthetic geometry data. Our findings highlight the benefits of certain model architectures and training techniques, including training with a data curriculum, the use of CNN-based visual encoders and the effect of LLM size and finetuning visual encoders. Of note, we find that adopting a data curriculum enables models to learn challenging geometric concepts that they fail to acquire from scratch. Finally, we explore how well we can endow multiple geometric low-level visual perception capabilities into one generalist MLLM. We demonstrate that training on such data with carefully chosen composition can significantly enhance a model's geometric visual perception ability *without compromising its general multimodal capabilities*, shedding light on the development of future generalist MLLMs that can excel simultaneously across multiple challenging domains.

## 1 INTRODUCTION

Multimodal large language models (MLLMs) have rapidly progressed in recent years, demonstrating remarkable potential in understanding and reasoning about the visual world through the powerful capabilities of LLMs (Liu et al., 2024c;a; OpenAI, 2025; Comanici et al., 2025; Bai et al., 2025). These models have showcased strong performance in tasks including visual question answering (VQA) (Goyal et al., 2017), image captioning (Lin et al., 2014), and multimodal reasoning (Yue et al., 2024; Jiang et al., 2024; Wang et al., 2024a; Hao et al., 2025).

While MLLMs achieve impressive results on various multimodal benchmarks, their performance often relies on high-level semantic extraction (Tong et al., 2024b); in contrast, they often fall short on *low-level visual perception* (LLVP)—*i.e.*, the ability to accurately describe the geometric details of an image, such as the points, lines, and spatial relationships among its constituent objects. This limitation becomes especially apparent in many tasks requiring precise visual descriptions, such as mathematical visual problem solving (Wang et al., 2024a), scientific visual understanding (Yue et al., 2024), spatial understanding for robotics (Chen et al., 2024a), medical image analysis for accurate diagnosis (Li et al., 2024b), GUI agents (Chen et al., 2024c), quality control in manufacturing, autonomous driving systems, augmented reality applications, and more.

In this paper, we choose 2D geometry as a focused testbed to study the challenges of low-level visual perception abilities in MLLMs, take steps to understand the root cause, and improve their performance. We start by developing *Geoperception*, a large scale *real-world* multimodal benchmark in plane geometry. Our findings reveal that most tasks from the benchmark are indeed challenging for current MLLMs. This limitation raises an important research question: *What is causing the difficulties exhibited by contemporary MLLMs in low-level visual perception tasks?* We hypothesize the main factor to be the insufficient availability of high-fidelity visual description datasets. Furthermore, in the absence of sufficient data, it is challenging to identify and adopt optimal architectural choices and training strategies.

To this end, we develop a synthetic dataset engine for large-scale generation of 2D geometry images paired with rich textual descriptions. This synthetic dataset enables us to conduct a *fully-controlled* empirical study exploring strategies for improving MLLMs' performance on low-level visual perception within the 2D geometry domain.

With our clean testbed, we offer new insights into MLLM training data and architectural choices from the perspective of low-level visual perception, deliberately excluding all high-level semantic understanding and text-based reasoning. Our empirical study reveals several *key lessons*. First, we find that incorporating simpler visual shapes early in the training process—a form of curriculum learning—enables the model to solve more complex perceptual tasks that it fails to learn from scratch. Moreover, our study highlights that for LLVP, scaling the LLM component provides negligible benefits, CNN-based visual encoders are more suitable than ViT architectures, and fine-tuning the vision encoder does not offer a strong advantage.

Finally, we explore how well we can endow multiple geometric low-level visual perception capabilities into one generalist MLLM. To this end, we construct *Euclid-200k*, a large-scale, diverse multimodal instruction-following dataset generated via our synthetic geometry data engine, and use it to train an off-the-shelf MLLM (Qwen-2.5-VL-3B (Bai et al., 2025)), resulting in a model that we refer to as Euclid. Experiment result shows that Euclid generalizes effectively in real-world geometric low-level visual perception, outperforming current leading MLLMs despite possessing only 3B parameters. Importantly, we further demonstrate that integrating Euclid-200k with general multimodal training datasets yields substantial gains in geometric low-level visual perception *without compromising broader multimodal perception and reasoning capabilities*. These findings shed light on the development of MLLMs that can excel simultaneously across multiple challenging domains.

## 2    BACKGROUND AND RELATED WORK

We provide an overview of prior efforts that assess and improve low-level perception and geometric reasoning in MLLMs, and highlight our contributions in data synthesis, evaluation, and training.

**Vision-Language MLLMs.**    While recent iterations of LLMs feature a standardized model architecture and pretraining recipe, MLLMs still often differ in design choices for infusing visual inputs. One popular design is to align *continuous* visual features with the embedding space of a backbone LLM (Liu et al., 2024b; Dubey et al., 2024; Tong et al., 2024a; AI, 2023; Bai et al., 2025); another approach involves *tokenizing* visual inputs to be trained jointly with language tokens (Team et al., 2023; Team, 2024a). These modules are often infused with a decoder-only LLM, but others have explored encoder-decoder architectures to integrate a more varied collection of modalities (Alayrac et al., 2022; Mizrahi et al., 2024; Ormazabal et al., 2024; Bachmann et al., 2024). Our study focuses on *decoder* MLLMs with a *continuous* visual encoder, and we carry out an empirical study to explore the effect of synthetic dataset mixture, training recipe, and encoder design (Liu et al., 2022; Radford et al., 2021; Zhai et al., 2023; Oquab et al., 2023).

**Geometry-Oriented MLLMs.**    At the core of these choices is the hardness in designing a module adept in general visual reasoning (McKinzie et al., 2024; Tong et al., 2024a). In this work, we explore the optimal design of MLLMs with emphasis on low-level visual perception, a crucial aspect for (among other applications) multimodal mathematical understanding (Lu et al., 2023; Zhang et al., 2024a). This paper supplements prior efforts in improving mathematical reasoning (Gao et al., 2023; Zhang et al., 2024b; Zhuang et al., 2024; Li et al., 2024c; Peng et al., 2024; Shi et al., 2024b) with a detailed study on the effect of dataset mixture, curriculum, and visual encoder, to reach a recipe that elicits strong performance on geometric tasks (Kazemi et al., 2023) that require low-level perception.

**Evaluating Low-level Visual Perception.**    Many benchmarks have reported that frontier-class MLLMs struggle with visual perception tasks (Rahmanzadehgervi et al., 2024; Fu et al., 2024b). These findings collectively identify that MLLMs exhibit a language prior (Lin et al., 2023)—a preference of textual inputs over visual inputs—leading to a performance gap between modalities (Wang et al., 2024c; Zhang et al., 2024a; Fu et al., 2024a). An LLVP-focused benchmark is recently proposed by Kamoi et al., which suggest that scaling LLM backbone improves LLVP performance. We show that careful selections of training curriculum and vision encoder can result in substantial performance gains, without scaling LLM backbone. Our findings are beneficial for practitioners with a low compute budget, or requiring deployment in edge devices with small models (Marafioti et al., 2025).

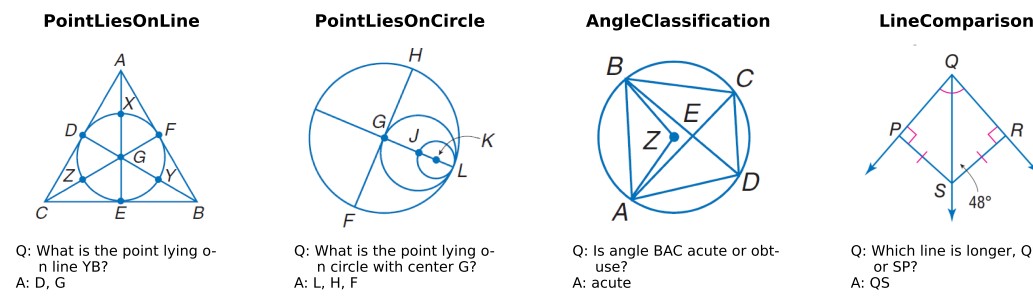

Figure 1: Four examples from Geoperception.

**Improving Low-level Visual Perception.** Many prior works study *data-driven* approaches to improve low-level perception skills. For example, Gao et al. (2023); Li et al. (2024c); Zhuang et al. (2024) employ a standardized supervised finetuning recipe, and optionally adjust the training data mixture. This type of training data is often synthesized from text-only math problems (Lu et al., 2021; Trinh et al., 2024) or via rule-based systems (Kazemi et al., 2023). In parallel, Vishniakov et al. (2023); Shi et al. (2024a); Tong et al. (2024b) have explored the design space of visual encoders for general-purpose vision-language reasoning. We identify best practices over the union of these design spaces, and then train small MLLMs with strong performance in low-level perception tasks.

## 3 GEOPERCEPTION

We begin our exploration by developing a benchmark to quantify current models' performance and to provide guidance on their generalization during development. To this end, we introduce *Geoperception*, a benchmark *dedicated* to low-level visual perception questions. The dataset is derived from real-world textbooks, ensuring *representativeness*, while also encompassing a wide variety of graph types, background colors, and annotation styles to maintain *diversity*.

**Benchmark Construction.** Over two thousand years ago, Euclid introduced five axioms that underpin all further geometric reasoning. These axioms involve establishing and extending lines using points (Axioms 1 and 2), constructing circles from a point and a radius (Axiom 3), and defining perpendicularity (Axiom 4) and parallelism (Axiom 5). Additionally, Euclid provided common notions regarding the properties of equality. To capture these aspects, we define five tasks in our Geoperception dataset: *PointLiesOnLine* (POL), *PointLiesOnCircle* (POC), *Parallel* (PRA), *Perpendicular* (PEP), *Equal* (EQL) and additionally define *AngleClassification* (ALC), and *LineComparison* (LHC) tasks to assess the model's understanding of angle and length measurements. Specifically, we define seven tasks: *PointLiesOnLine* (POL), *PointLiesOnCircle* (POC), *Parallel* (PRA), *Perpendicular* (PEP), *Equal* (EQL), *AngleClassification* (ALC), and *LineComparison* (LHC). In geometric diagrams, perpendicularity, parallelism, and equality are often denoted by annotation symbols; thus, these tasks are categorized as geometry annotation understanding. In contrast, *PointLiesOnLine*, *PointLiesOnCircle*, *AngleClassification*, and *LineComparison* fall under primitive geometric shape understanding, which further divides into logical (POL, POC) and numerical (ALC, LHC) tasks. We leverage the precisely annotated logical forms for geometric diagrams provided by Geometry-3k (Lu et al., 2021) to construct Geoperception. Additional construction details are provided in Appendix B.

**Evaluation Details.** We evaluate fifteen leading MLLMs, both open source and closed source. The open source models include Molmo-7B-D (Deitke et al., 2024), Cambrian-1-8B (Tong et al., 2024a), Llama-3.2-11B (Dubey et al., 2024), Pixtral-12B (AI, 2023) and Qwen-2.5-VL-72B (Bai et al., 2025). The closed-source models include GPT-family (Achiam et al., 2023; OpenAI, 2025), Claude-family (Anthropic, 2024; 2025), Gemini-family (Team et al., 2023; Comanici et al., 2025). We use carefully designed prompts to specify the model's output format (detailed in Appendix A) and define our *evaluation score* to be 1 if $P \subseteq G$ and 0 otherwise, where $G$ denotes the ground truth set of answers (a list of points, lines), and $P$ denotes the prediction set of answers. GPT-4o-mini without image input is used for generating the random baseline for both benchmarks, employing the same textual instructions. Additionally, we randomly sample 50 question to test human performance on out task. Results are shown in Table 1, and full prompts can be found in Appendix A.

Table 1: Performance (avg. evaluation score) of different models on Geoperception benchmark tasks. POL: PointLiesOnLine, POC: PointLiesOnCircle, ALC: AngleClassification, LHC: LineComparison, PEP: Perpendicular, PRA: Parallel, EQL: Equals. As the Random Baseline method, we use GPT-4o-mini, given the same textual instruction but without an image. The best model for each task is **bolded**.

| | Logical | | Numerical | | Annotations | | | |
| Model | POL | POC | ALC | LHC | PEP | PRA | EQL | Overall |
| --- | --- | --- | --- | --- | --- | --- | --- | --- |
| Random Baseline | 1.35 | 2.63 | 59.92 | 51.36 | 0.23 | 0.00 | 0.02 | 16.50 |
| Human | 100.00 | 98.00 | 98.00 | 96.00 | 98.00 | 100.00 | 90.00 | 97.14 |
| *Open Source* | | | | | | | | |
| Molmo-7B-D (Deitke et al., 2024) | 11.96 | 35.73 | 56.77 | 16.79 | 1.06 | 0.00 | 0.81 | 17.59 |
| Llama-3.2-11B (Dubey et al., 2024) | 16.22 | 37.12 | 59.46 | 52.08 | 8.38 | 22.41 | 49.86 | 35.08 |
| Cambrian-1-8B (Tong et al., 2024a) | 15.14 | 28.68 | 58.05 | 61.48 | 22.96 | 30.74 | 31.04 | 35.44 |
| Pixtral-12B (AI, 2023) | 24.63 | 53.21 | 47.33 | 51.43 | 21.96 | 36.64 | 58.41 | 41.95 |
| Qwen-2.5-VL-72B (Bai et al., 2025) | 50.39 | 72.42 | 56.27 | 72.24 | 33.36 | 68.87 | 60.32 | 59.13 |
| *Closed Source* | | | | | | | | |
| GPT-4o-mini (Achiam et al., 2023) | 9.80 | 61.19 | 48.84 | 69.51 | 9.80 | 4.25 | 44.74 | 35.45 |
| GPT-4o (Achiam et al., 2023) | 16.43 | 71.49 | 55.63 | 74.39 | 24.80 | 60.30 | 44.69 | 49.68 |
| GPT-4.1 (OpenAI, 2025) | 55.02 | 77.99 | 58.41 | 79.70 | 40.14 | 87.74 | 72.25 | 67.32 |
| GPT-5-mini (OpenAI, 2025) | 60.39 | 81.34 | 53.85 | 73.89 | 42.67 | 58.49 | 54.98 | 60.80 |
| GPT-5 (OpenAI, 2025) | 40.72 | 81.06 | 55.49 | 79.77 | 33.20 | 76.42 | 63.98 | 61.52 |
| Claude 3.5 Sonnet (Anthropic, 2024) | 25.44 | 68.34 | 42.95 | 70.73 | 21.41 | 63.92 | 66.34 | 51.30 |
| Claude 4 Sonnet (Anthropic, 2025) | 55.18 | 81.34 | 67.31 | 79.99 | 53.55 | 87.74 | 73.92 | 71.29 |
| Gemini-1.5-Flash (Team et al., 2023) | 29.30 | 67.75 | 49.89 | 76.69 | 29.98 | 63.44 | 66.28 | 54.76 |
| Gemini-1.5-Pro (Team et al., 2023) | 24.42 | 69.80 | 57.96 | 79.05 | 38.81 | 76.65 | 52.15 | 56.98 |
| Gemini-2.5-Flash (Comanici et al., 2025) | 44.56 | 84.12 | 65.62 | 80.92 | 62.85 | 93.40 | 65.67 | 71.02 |
| Gemini-2.5-Pro (Comanici et al., 2025) | **68.17** | **86.35** | **78.84** | **84.58** | **77.13** | **97.17** | **75.92** | **81.17** |

**Current MLLMs Struggle to Perceive Geometric Low-level Visual Details.** Overall, there remains a substantial gap between human and model performance, with the best-performing MLLM (Claude 4 Sonnet) trailing humans by 26%. Among the tasks, models tend to underperform on primitive geometric tasks. Notably, all evaluated models fail to outperform 10% above the random baseline on the angle classification task, which simply requires identifying whether an angle is acute or obtuse—excluding near-right angles in the 80–100 degree range. In contrast, models perform better on annotation-recognition tasks. Gemini-2.5-Flash achieves over 90% accuracy in recognizing parallel lines, while Claude 4 Sonnet exceeds 73% accuracy in identifying equal lines and angles. Lastly, it is worth noting that Cambrian-1 (Tong et al., 2024a), which is trained on Geo-170k (Gao et al., 2023), a geometry multimodal instruction tuning dataset built on the logical annotation of Geometry-3k (the same source as Geoperception), still faces challenges in our Geoperception task, despite being trained on the dataset having the same images and augmented text annotations.

## 4  WHAT IS CAUSING LLVP SHORTCOMINGS IN MLLMS?

We next aim to investigate *what is causing such LLVP shortcomings in MLLMs?* We hypothesize that insufficient multimodal datasets with detailed low-level visual descriptions, or sub-optimal model design choices (Tong et al., 2024b), may be key reasons. However, in the absence of sufficient datasets, it is challenging to identify and adopt architectural choices and training strategies that could improve LLVP-specific training. Hence, our investigation starts by training MLLMs on a proposed dataset with rich low-level visual description, followed by a detailed ablation of MLLM design space.

**Synthetic 2D Geometry as a Focused Testbed.** Without the knowledge of how MLLMs can learn low-level visual features, it is challenging to adopt real-world dataset for the investigation, since they are uncontrollable and often contains limited low-level visual descriptions. Hence, we opt to use *controllable* and *scalable* synthetic data for our investigation and choose 2D geometry as a focused testbed due to its rich low-level visual features. We develop a geometry image generation engine which accepts textual representations of geometry shapes and is able to render virtually infinite geometry images. The full detail of the dataset engine is described in Section 5. For our empirical study, we define three geometry shape textual representations and two low-level visual tasks *PointLiesOnLine*, where the model is given two points and asked for all other points on the same line and *LineComparison*, where the model is asked to compare the length of two lines.

**Instruction-Tuning MLLMs from Scratch for a Clean Setting.** While many off-the-shelf MLLMs are readily available, their training datasets are often highly complex or unavailable, making it difficult to isolate the impact of different dataset sources. Furthermore, it is also impractical to separately study the impact of different MLLM components by comparing existing MLLMs,

since they vary substantially in architectural components (e.g., vision encoder, language model, and multimodal connector), training data, and hyperparameter configurations. Hence, we opt to finetune MLLMs from scratch with off-the-shelf visual encoders and LLMs.

**Empirical Investigation.** We begin by adopting the most widely used architecture in current MLLMs (Liu et al., 2024c), which comprises a ViT visual encoder (we use CLIP-ViT-L/14 (Radford et al., 2021)), a LLM (we use Qwen-2.5 series (Team, 2024b)), and a lightweight MLP connector bridging the two modalities. In our setup, the visual encoder is kept frozen, while both the MLP and the LLM are actively tuned during training. To simplify the training process, we employ geometry images generated by our rendering engine and pair them with templated question-answer instances.

### 4.1 HOW DATASET COMPOSITION AFFECTS LLVP PERFORMANCE

We train the model on two tasks: *PointLiesOnLine* where the model learns the relationship between points and lines; and *LineComparison*, where the model learns to compare the length of two segments. Each task is under three different geometry problem difficulty levels controlled by the complexity of geometry shapes, with each configuration using a dataset of 32k samples (corresponding to 500 steps with a batch size of 64)—a substantial volume for a single-task setting. To ensure robustness, each experiment is repeated three times, and we report the in-domain performance after training for each set of experiment. The result is shown in Fig. 2a. We observe that the model achieves high accuracy for the *Easy* difficulty problems, while it struggles for other more-complex geometry problems (*Medium* and *Hard*). These result indicate that simply increased data volume alone is insufficient to fully address the difficulty of more-complex LLVP tasks, which remain challenging.

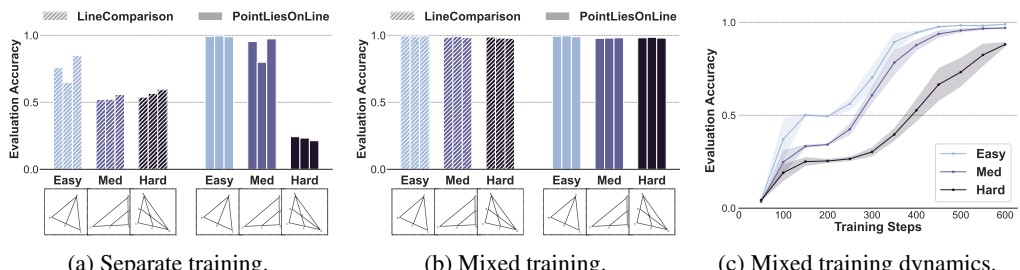

|                        |                     |                            |
| :--------------------: | :-----------------: | :------------------------: |
| (a) Separate training. | (b) Mixed training. | (c) Mixed training dynamics. |

Figure 2: Illustration of training on different data compositions. **(a)** We train the model on 32k examples separately on each problem difficulty level and report the evaluation accuracy on in-domain test points. **(b)** We train the model on 96k examples on the mixture of three problem difficulty levels and report the accuracy on each level separately. **(c)** In *PointLiesOnLine* task training on the mixture of three levels, we separately report the evaluation accuracy on each level, with respect to the number of training steps. In these plots, each bar represents one of three runs in an experiment.

In Fig. 2a, we observe that the model attains higher performance on relatively simple geometric problems but struggles to converge on more complex ones. We then investigate whether early convergence on simpler problems can facilitate the learning of the same visual concept in more complex contexts. To this end, we construct a mixed training set comprising all three problem difficulty levels. For fairness, we keep the per-task data volume the same as in the previous setting (32k samples), resulting in 96k training samples in total. As shown in Fig. 2b, joint training across all problem difficulty levels enables the model to achieve near-perfect accuracy even on complex shapes, which it failed to do previously. This might suggest that the presence of simpler shapes in training can aid in learning more difficult structures[1].

To test this hypothesis, we examine the testing accuracy for each difficulty level separately during mixed training on the *PointLiesOnLine* task, as shown in Fig. 2c. We observe that accuracy for simpler shapes rises earlier and more rapidly than that for complex ones. Furthermore, the performance on the most complex shapes plateaus during the initial 100–250 training steps, while the accuracy on the two simpler shapes continues to improve. These findings support the view that early convergence on simpler low-level visual patterns can substantially benefit the learning of more challenging ones, aligning with the principles of curriculum learning. This leads us to our first lesson for LLVP training:

---

[1]Interestingly, on *LineComparison* task, the inclusion of harder shapes in the training process also enhances the model's performance on simpler ones, suggesting a mutual benefit.

**Lesson 1:** Exposure to simpler visual perception data aids in learning difficult LLVP.

Overall, our results indicate that, unlike humans, MLLMs do not acquire robust low-level visual perception—arguably a vision-centric ability—merely through large-scale vision-language pretraining. Instead, such abilities require carefully designed data compositions and targeted training strategies. We next investigate how architectural choices in MLLMs influence this behavior.

## 4.2 HOW MODEL ARCHITECTURE CHOICES AFFECT LLVP PERFORMANCE

In this section, we examine several representative architectural design choices commonly adopted in contemporary MLLMs. To ensure controlled comparisons, we conduct a series of ablation experiments in which each design choice is individually varied while keeping all other components fixed. All models are trained from scratch on the same dataset—a mixture of three geometric shapes as used in Fig. 2. To assess performance, we report both the training loss and testing accuracy.

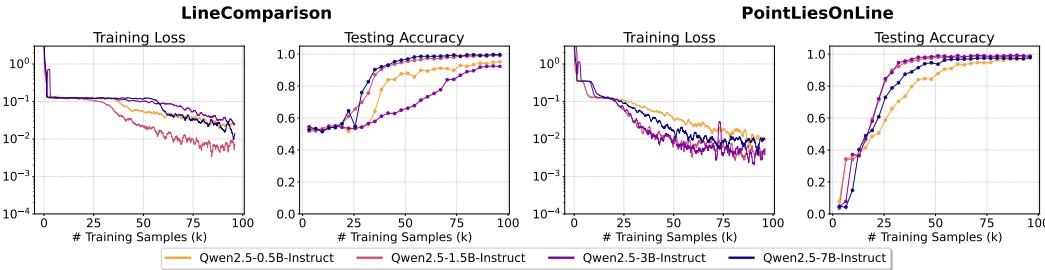

Figure 3: LLM size experiments. We show training loss and test accuracy (on a 1500-instance holdout set), comparing four choices of LLM size with a fixed visual encoder and multimodal connector. Training losses are window-smoothed using a window size of 10 for better visibility.

Reflected in most MLLM releases (Liu et al., 2024a; Tong et al., 2024a; Wang et al., 2024b), scaling up the LLM often results in improved MLLM performance on standard benchmarks, when trained on the same dataset. However, since their training datasets are complex mixtures from multiple sources, it is unclear whether this improvement stems from enhanced language-space reasoning or better visual perception. In contrast, our training dataset is designed with simplicity in the language space (following specific templates) and focuses on low-level visual perception abilities. This setup allows us to better isolate and analyze the source of performance gains, providing clearer insights into the ability to learn better low-level visual perception.

We use four variants of Qwen-2.5 (Team, 2024b): 0.5B, 1.5B, 3B and 7B, while keeping other components in the MLLMs consistent and training them on the same dataset. The results are shown in Fig. 3. First, we observe a sharp decrease in loss at the start of training, which corresponds to the LLM adapting to answer templates, indicating no significant difficulty across different model sizes. The subsequent loss decrease after the plateau marks the beginning of learning low-level visual features, as also evidenced by the testing accuracy. For *LineComparison*, Qwen-2.5-1.5B performs the best, while Qwen-2.5-3B learns most slowly. For *PointLiesOnLine*, Qwen-2.5-1.5B and Qwen-2.5-3B perform nearly identically. while Qwen-2.5-0.5B learns relatively slower but eventually reaches a similar final performance as the other models. In summary, we do not observe a clear trend that larger LLMs learn low-level visual perception tasks faster or better[2]. Based on these findings, we will use Qwen-2.5-1.5B for further exploration.

**Lesson 2:** Scaling LLM size yields limited gains in LLVP learning.

We then study the choice of visual encoder architectures, including two families of architectures: Vision Transformer (ViT) (Dosovitskiy, 2020) and ConvNeXT (Liu et al., 2022); as well as two

---

[2]The BLINK (Fu et al., 2024b) benchmark shares similar observations. For example, among 14 tasks, LLaVA-1.5-13B outperforms its 7B variant in only 4 tasks. Additionally, LLaVA-one-vision (Li et al., 2024a)'s 0.5B variant outperforms its 7B variant by 3.9% and underperforms its 72B variant by only 3.3%.

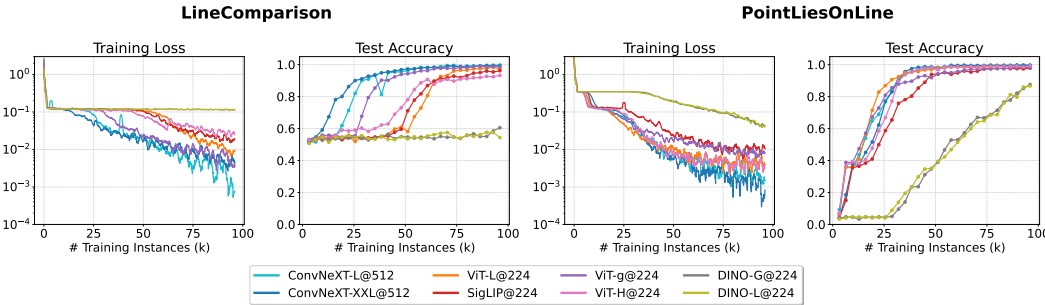

Figure 4: Vision encoder experiments. We show training loss and test accuracy (on a 1500-instance holdout set), comparing eight visual encoders, with a fixed multimodal connector and LLM. For a fair comparison, all visual encoder transcribe an image into 256 visual tokens. Training losses are window-smoothed using a window size of 10 for better visibility.

visual representation learning objectives: language-supervised learning (Radford et al., 2021) and self-supervised learning (Oquab et al., 2023). We control the number of visual tokens to 256 for all of our vision encoders, a summary of all visual encoders can be found in Table 5. The result is shown in Fig. 4. We find that ConvNeXt-XXLarge and ConvNeXt-Large consistently learns the fastest among all of the visual encoders. Notably, ConvNeXT-Large shows superior learning performance with the vision transformers which are 3-5 times larger. We hypothesize that CNN architecture extract visual features globally, effectively preserving low-level visual features. In contrast, ViT architectures split images into discrete patches, making it more challenging to retain the original low-level visual information. That being said, nearly all visual encoders (except DINO-v2) reaches near perfect performance after trained on sufficient datasets. Self-supervised learning (SSL) visual encoders, DINO-v2, struggles to learn the geometry concept; we hypothesis this is due to the weak vision-language representation in these models. Surprisingly, although the SigLIP-family is widely-recognized as a better visual encoder (Tong et al., 2024a; Li et al., 2024a), we find that their performance in learning basic visual geometry attributes is limited.

> **Lesson 3:** CNN-based visual encoders enable more efficient learning of LLVP tasks.

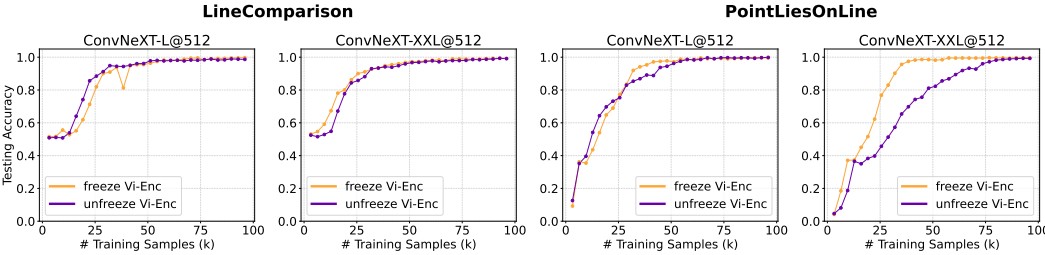

Figure 5: Tuning/freezing vision encoder experiments. We show test accuracy (on a 1500-instance holdout set), comparing freezing versus tuning the visual encoder during training.

Actively tuning the visual encoder has the potential to learn better visual representations, which could be helpful for LLVP. We aim to empirically assess the effect of tuning versus freezing the visual encoder. In Fig. 5, we show the testing accuracy curves of tuning and freezing visual encoders. Surprisingly, we find that compared with using a frozen encoder, tuning the visual encoder does not help the model learn LLVP faster or better. This suggests that current visual encoders seem to be able to preserve adequate low-level visual information, and it's sufficient to just train LLMs to make better use of the visual features.

> **Lesson 4:** Tuning the vision encoder does not offer a strong advantage in learning LLVP.

## 5 EUCLID: ENDOWING GENERAL MLLM WITH STRONG GEOMETRIC LLVP

In this section, we explore how well we can endow multiple geometric low-level visual perception capabilities into a generalist MLLM. To this end, we first expand the synthetic dataset diversity and quantity and use it to train an off-the-shelf MLLM, Qwen-2.5-VL-3B (Bai et al., 2025), resulting in a model named Euclid. Next, we will introduce our geometry dataset generation engine in full, together with our resulting dataset statistics.

### 5.1 GEOMETRY DATASET ENGINE

Our geometry dataset generation engine is designed to be able to produce virtually infinite synthetic perception instances. The data generation process consists of three steps: geometry logical shape sampling, data instance rendering, and training dataset generation.[3]

**Geometry Logical Shape Sampling.** A geometry logical shape refers to a textual sequence of geometry constructions that collectively define a geometric configuration (*e.g.*, a right triangle, followed by a midline from the right vertex). Each construction step can be categorized as free (*e.g.*, constructing an arbitrary segment), half-free (*e.g.*, placing a random point on a given line), or fully constrained (*e.g.*, constructing the intersection of two predefined segments). We adopt the construction primitives defined in AlphaGeometry (Trinh et al., 2024) and use a refined set of 41 construction types. During sampling, our engine dynamically enumerates all valid constructions permitted by the current state and randomly selects a new one. Both the sequence length and the distribution over construction types are fully controllable. Using this procedure, we generate 10,000 unique geometry logical shapes, distributed as: 2,000 shapes of length 2, 6,000 of length 3, 1,000 of length 4, and 1,000 of length 5. A subset of geometry logical shapes is presented in Fig. 7

**Data Instance Rendering.** For each logical shape, the engine attempts to render a corresponding geometry image on a canvas. It outputs the geometry image along with a detailed metadata record describing the instance. This metadata includes numerical properties—such as the relative (unified, from 0-1) coordinates of all points, segment lengths, and angle measure—as well as logical relationships including collinearity, cyclicity, perpendicularity, parallelism, and equality of lengths/angles. For each logical shape, we render roughly 20 instances, resulting in 199,568 image-metadata pairs.

**Training Dataset Generation.** To convert each rendered geometry instance into a set of natural language question-answer pairs, we leverage a semi-automatic pipeline, where we use Gemini-2.5-Pro (Comanici et al., 2025) to generate 100 templates for each question type, and a template is randomly sampled to generate each question and answer. This semi-automatic manner ensures the diversity and accuracy of the dataset generated. We call the resulting dataset *Euclid-200k*.

### 5.2 MODEL TRAINING AND EVALUATION

We train an off-the-shelf state-of-the-art MLLM, Qwen-2.5-VL-3B on Euclid-200k, with batch-size of 128 by LLaMA-Factory (Zheng et al., 2024), on a single H100 GPU. We evaluate our model from multiple aspects, including Geoperception, as well as several general multimodal tasks.

**Evaluation on Geoperception.** The results are detailed in Table 2. Despite being exclusively trained on a synthetic geometry dataset and possessing only 3B parameters, Euclid significantly outperforms current leading MLLMs on most Geoperception tasks from the real world, showing strong generalization abilities on real-world geometry images. Notably, on the *PointLiesOnLine* task, a particularly challenging evaluation for existing MLLMs, Euclid achieves an accuracy of 79.43%, surpassing Claude 4 Sonnet by 25%. Notably, in the numerical tasks—*LineComparison* and *AngleClassification*—Euclid attains exceptionally high accuracy rates of 98.13% and 92.43%, respectively, approaching perfect performance. Additionally, Euclid yields consistent gains over both its baseline and much larger variants, Qwen-2.5-VL-3B and Qwen-2.5-VL-72B, despite being post-trained on fewer than 200k samples. This highlights the decisive role of dataset quality in enhancing performance on low-level visual perception tasks and demonstrates that, with carefully

---

[3]It is worth noting that our synthetic training dataset for Euclid is fundamentally distinct from the Geoperception benchmark, which is derived from real textbook diagrams including various diagram shapes, manual annotations and styles that are not producible by our dataset engine.

Table 2: Performance comparison between Euclid, it based model (Qwen-2.5-VL-3B) and the best leading MLLMs on the seven Geoperception tasks. Note that Euclid is *not* trained on any of the in-distribution data from the benchmark tasks below. The best model for each task is **bolded**.

| Model | Logical | | Numerical | | Annotations | | | Average |
|---|---|---|---|---|---|---|---|---|
| | POL | POC | ALC | LHC | PEP | PRA | EQL | |
| Random Baseline | 0.43 | 2.63 | 59.92 | 51.36 | 0.25 | 0.00 | 0.02 | 16.37 |
| Qwen-2.5-VL-3B (Bai et al., 2025) | 15.36 | 18.94 | 23.33 | 16.64 | 11.32 | 50.89 | 51.72 | 26.89 |
| Qwen-2.5-VL-72B (Bai et al., 2025) | 50.39 | 72.42 | 56.27 | 72.24 | 33.36 | 68.87 | 60.32 | 59.13 |
| Claude 4 Sonnet (Anthropic, 2025) | 55.18 | 81.34 | 67.31 | 79.99 | 53.55 | 87.74 | 73.92 | 71.29 |
| Gemini-2.5-Pro (Comanici et al., 2025) | 68.17 | **86.35** | 78.84 | 84.58 | **77.13** | **97.17** | **75.92** | **81.17** |
| Euclid | **79.43** | 74.37 | **92.43** | **98.13** | 68.06 | 76.42 | 62.08 | 78.70 |

curated data, smaller models can surpass considerably larger ones. Nevertheless, in annotation tasks, although Euclid attains competitive results, it does not consistently outperform all models. We hypothesize this limitation arises from the inherent complexity and diversity of annotation types present in Geoperception, which originate from authentic educational materials.

Table 3: Performance comparison between Euclid*, Euclid, and their baselines across a broad set of benchmarks. Full evaluation details are provided in Appendix A. Random baseline is reported for the Geoperception benchmark introduced in this paper.

| Model | Geoperception | MMBench | MMStar | MathVista | Math-Vision | We-Math | GeoQA |
|---|---|---|---|---|---|---|---|
| Random Baseline | 16.37 | 2.90 | 24.60 | 17.90 | 5.86 | 21.59 | 25.00 |
| Qwen-2.5-VL-3B | 26.89 | 77.84 | 47.87 | **52.08** | 15.92 | 35.34 | 51.48 |
| Qwen-General-Only | 30.00 | 81.01 | 47.73 | 51.66 | **16.79** | 38.33 | 53.43 |
| Euclid | **78.70** | 74.87 | 43.20 | 50.23 | 13.87 | 31.90 | 38.26 |
| Euclid* | 78.53 | **81.52** | **48.13** | 51.38 | **16.79** | **42.13** | **61.66** |

**Compatibility with General-Purpose Scenarios.** We further examine whether the large-scale synthetic Euclid-200k dataset is compatible with general-purpose scenarios. To this end, we integrate Euclid with a selection of diverse real-world datasets, aiming to maintain the model's ability to follow general instructions and understand a wide range of visual contexts. Specifically, we incorporate subsets from LLaVA-One-Vision (Li et al., 2024a), comprising four math reasoning datasets: 15k Mavis-meta-gen samples, 15k Mavis-rule-geo samples (Zhang et al., 2024b), 15k G-LLaVA alignment samples, 15k G-LLaVA QA samples (Gao et al., 2023), and 30k general visual instruction tuning instances from the Cambrian-1 dataset (Tong et al., 2024a). The combined dataset is utilized to train a model we designate as Euclid*. Additionally, we train another model solely on the aforementioned general-purpose datasets, termed Qwen-General-Only, as a baseline for ablation analysis.

We evaluate Euclid, Euclid*, Qwen-General-Only, and the base Qwen-2.5-VL-3B model on multiple real-world multimodal benchmarks, including MMBench (Liu et al., 2023), MMStar Chen et al. (2024b), MathVista (Lu et al., 2023), Math-vision (Wang et al., 2024a), We-Math (Qiao et al., 2025) and GeoQA (Chen et al., 2021), the evaluation result is shown in Table 3. We find that *with careful dataset composition, large-scale synthetic data can significantly improve MLLMs' performance on low-level visual perception and geometry math reasoning tasks without compromising their general multimodal capabilities.* Notably, despite being entirely synthetic and focused on plane geometry, Euclid-200k allows for measurable gains on real-world benchmarks when integrated with general-purpose data. For instance, when combined with subsets from diverse instruction-following datasets, the resulting model surpasses Qwen-2.5-VL-3B by 10.18% on GeoQA, 6.69% on We-Math and 3.68% on MMBench. These findings suggest that, high quality synthetic datasets like Euclid-200k can complement real-world data and contribute meaningfully to the overall performance of generalist MLLMs. Such finding is particularly noteworthy as future applications may require MLLMs to acquire increasingly domain-specific capabilities; ensuring that such specialization does not undermine their general abilities highlights the potential of this approach for developing robust and versatile models.

## 6 CONCLUSION

In this work, we identify geometric low-level visual perception as a core limitation of current MLLMs by assessing it through a dedicated benchmark named Geoperception. Then we present a comprehensive empirical study with fully controlled experiment that investigate strategies for improving model's performance, resulting in four lessons that highlight the benefits of certain model architectures and training techniques. We also explore how well we can endow multiple geometric LLVP into a general MLLM. Importantly, we show that general multimodal capabilities can be preserved when synthetic training data is complemented with a modest portion of general-domain multimodal data. Looking ahead, extending progress on low-level visual perception in general domains is a promising, crucial yet challenging direction for the development of MLLMs.

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

# APPENDIX

## A  EVALUATION DETAILS

---

**PROMPT TEMPLATE FOR GPT-4O-MINI TO PRODUCE RANDOM BASELINE**

```
You are given a question that is usually accompanied by an image, but no
image is provided here.
Based only on the question text, give the most likely correct answer.
Do not mention the absence of the image.
Do not use phrases like 'I guess', 'I think', 'it might be', or anything
uncertain.
Just provide a direct and confident answer.
```

---

**PROMPT TEMPLATE FOR THE POINTLIESONLINE TASK**

```
Answer directly just with all points lying on the line mentioned in the
question (do not include the point mentioned in the question).
Answer template:  (if there is only one point) The other point is:
"your_point".  Or (if there are multiple points) The other points are:
"your_points".
For example:  The another point is:  A
Or:  The other points are:  A, B, C
```

---

**PROMPT TEMPLATE FOR THE POINTLIESONCIRCLE TASK**

```
Answer directly just with all points lying on the circle mentioned in the
question.
Answer template:  (if there is only one point) The point is:  "your_point".
Or (if there are multiple points) The points are:  "your_points".
For example:  The point is:  A
Or:  The points are:  A, B, C
```

---

**PROMPT TEMPLATE FOR THE PERPENDICULAR TASK**

```
Answer directly just with all lines which are perpendicular to the line
mentioned in the question (do not include the line mentioned in the
question).
Answer template:  (if there is only one line) The line is:  "your_line".  Or
(if there are multiple lines) The lines are:  "your_lines".
For example:  The line is:  BC
Or:  The lines are:  BC, DE
```

**PROMPT TEMPLATE FOR THE PARALLEL TASK**

```
Answer directly just with all lines which are parallel to the line mentioned
in the question (do not include the line mentioned in the question).
Answer template:  (if there is only one line) The line is:  "your_line".  Or
(if there are multiple lines) The lines are:  "your_lines".
For example:  The line is:  BC
Or:  The lines are:  BC, DE
```

**PROMPT TEMPLATE FOR THE EQUALS TASK**

```
Answer directly just with the annotations presented on the image.
Answer template:  The annotation is:  "your_annotation".
For example:  The annotation is:  2x+4
Or:  The annotations is:  90
```

**PROMPT TEMPLATE FOR THE LINECOMPARISON TASK**

```
Answer directly just with the longer line mentioned in the question.
Answer template:  The longer line is:  "your_line".
For example:  The longer line is:  BC
Or:  The longer line is:  DE
```

**PROMPT TEMPLATE FOR THE ANGLE CLASSIFICATION TASK**

```
Answer directly just with the classification of the angle mentioned in the
question.
Answer template:  The angle is:  "your_angle".
For example:  The angle is:  acute
Or:  The angle is:  obtuse
```

## B GEOPERCEPTION BENCHMARK DETAILS

**Data Filtering.** Geoperception is sourced from the Geometry-3k (Lu et al., 2021) corpus, which offers precise logical forms for geometric diagrams, compiled from popular high-school textbooks. However, certain points in these logical forms are absent in the corresponding diagrams. To resolve this, we use GPT-4o-mini MLLM to confirm the presence of all points listed in the logical forms. This process filters the 3,002 diagrams to retain 1,584, where at least one logical form fully represents its points in the diagram. A random inspection of 100 annotations reveals only two errors, indicating high annotation accuracy.

**Converting Logical Forms Into Questions.** We convert logical forms into question-and-answer pairs for each of the seven tasks in Geoperception. In the `Equals` task, for example, we directly convert the logical form (e.g., `Equals(LengthOf(Line(Q, T)), 86)`) into a question-answer pair (e.g., `Q: What is the length of line QT as annotated? A: 86`). For `PointLiesOnLine`, two points on the line are chosen to form the question, with the remaining points on the line as the answer. Similarly, for `PointLiesOnCircle`, we ask which points lie on the circle, using its center as the basis for the question. For `Parallel` and `Perpendicular`, we represent each line by two points and query which other lines are parallel or perpendicular to it. In `AngleClassification`, we ensure the queried angle is in the range of $[10, 80] \cup [100, 170]$ degrees to avoid ambiguity. For `LineComparison`, we ensure that the shorter line is less than 70% of the length of the longer line. Since multiple equivalent questions can be generated for a single logical form (e.g., a line containing five points generates $^5P_2$ equivalent questions), we randomly select one to avoid redundancy. Table 4 summarizes the question statistics for each task, as well as the number of images involved. Extended examples from Geoperception are illustrated in Fig. 6, where we show four examples for each of the question category.

**Statistics.** In Table 4, we provide more details on the Geoperception benchmark, such as the number of logic forms present before and after filtering, the number of questions, and the number of images. `AngleClassification` and `LineComparison` are directly derived from points coordinates without filtering.

| Predicate | # LF Before Filter | # LF After Filter | # Q | # I |
|---|---|---|---|---|
| PointLiesOnLine | 6988 | 2567 | 1901 | 924 |
| PointLiesOnCircle | 1966 | 1240 | 359 | 322 |
| Parallel | 222 | 123 | 106 | 101 |
| Perpendicular | 1111 | 680 | 1266 | 456 |
| Equals | 6434 | 4123 | 4436 | 1202 |
| AngleClassification | - | - | 2193 | 1389 |
| LineComparison | - | - | 1394 | 1394 |

Table 4: Statistics of the five predicates in our Geoperception dataset. Including number of logic forms before filter, after filter and the number of questions and images.

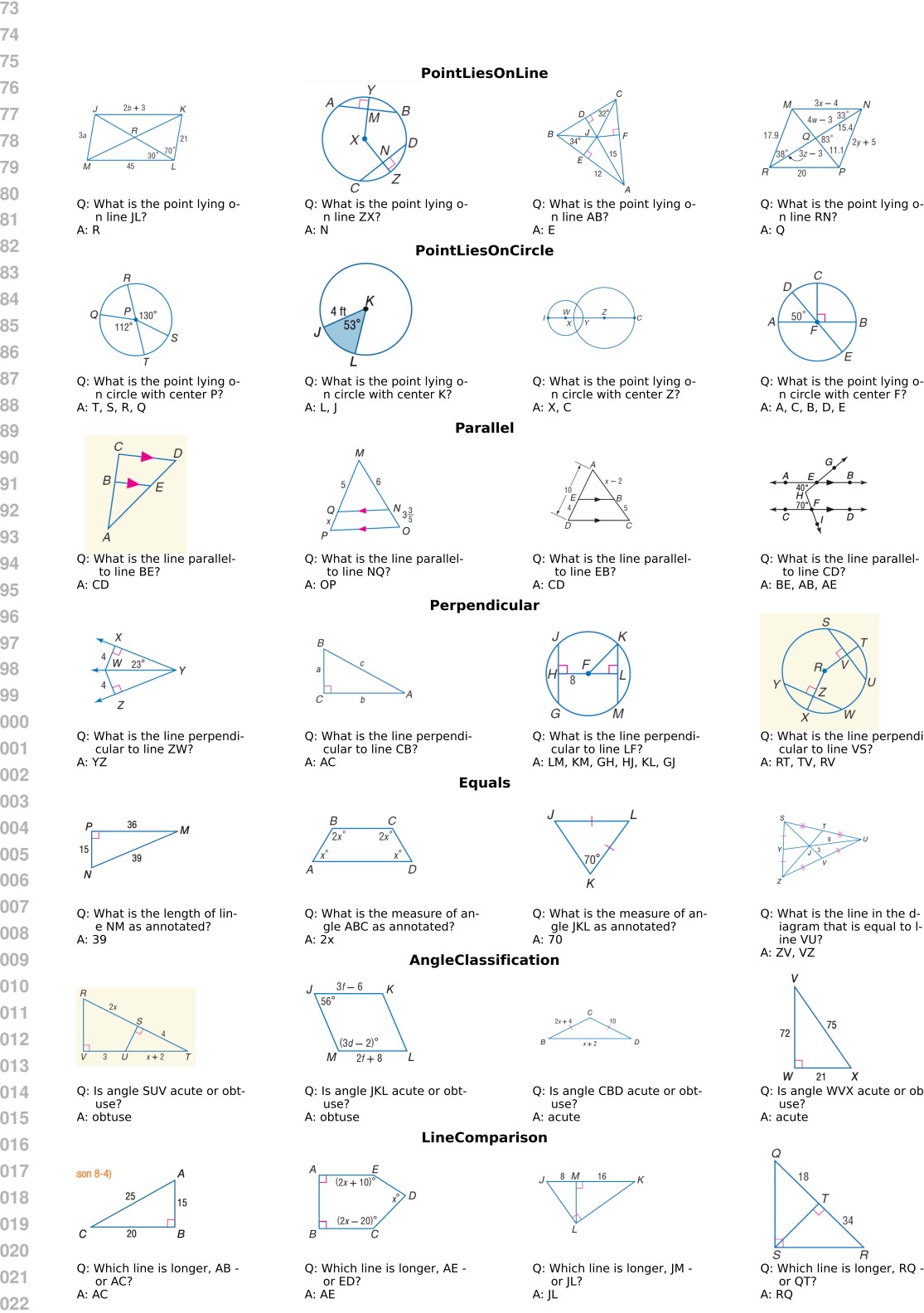

Figure 6: Examples of our Geoperception dataset.

## C Examples of Euclid-200k Dataset.

In Fig. 7, we randomly sample 50 geometry shapes from our 10k geometry logical shapes, which are used to produce Euclid-200k dataset. In Fig. 8, we randomly sample 192 images from Euclid-200k dataset.

---

**GEOMETRY SHAPE GENERATION CODE**

```
A B C = risos A B C; D E = trisegment D E C B
A B C = r_triangle A B C; D = incenter D A C B
A B C D = eq_trapezoid A B C D; E = shift E C D B
A B C D = quadrangle A B C D; E F G H = trapezoid E F G H
A B C = triangle12 A B C; D = on_tline D A B C, on_tline D B A C
A B C D = eq_trapezoid A B C D; E = free E; F = incenter F B E C
A B C = ieq_triangle A B C; D = on_bline D A C, on_tline D A B A
A B C = r_triangle A B C; D = lc_tangent D C B, on_tline D B C A
A B C = iso_triangle A B C; D = eqdistance D B A C, on_line D C B
A B C = ieq_triangle A B C; D = eqdistance D C B C, on_bline D C B
A B C = triangle A B C; D E F = r_triangle D E F; G = foot G F A E
A B C D = r_trapezoid A B C D; E = eqdistance E C D B, on_bline E A C
A B C D = eq_quadrangle A B C D; E = on_pline E A D C, on_tline E C D B
A B C D = trapezoid A B C D; E F G = triangle12 E F G; H = midpoint H B F
A B C D = eq_quadrangle A B C D; E = eqdistance E B C B, on_pline E D A B
A B C = risos A B C; D E F G = quadrangle D E F G; H I = trisect H I G F B
A B C D = eq_trapezoid A B C D; E = angle_bisector E C D B, on_bline E C A
A B C = ieq_triangle A B C; D E F G = rectangle D E F G; H = circle H G C B
A B C D = rectangle A B C D; E F G H = isquare E F G H; I = incenter I F E A
A B C = triangle12 A B C; D = on_bline D C A, on_dia D C A; E = on_bline E C B
A B C D = rectangle A B C D; E = on_bline E D A, on_line E B D; F = foot F E A D
A B C D = trapezoid A B C D; E F G H = eq_quadrangle E F G H; I = excenter I A G H
A B C = risos A B C; D = eqdistance D B A C, on_tline D A C A; E F = trisegment E F C A
A B C D = eqdia_quadrangle A B C D; E F G H = eqdia_quadrangle E F G H; I = foot I H F E
A B C D = rectangle A B C D; E = eqdistance E B C B, on_pline E A C D; F = midpoint F E A
A B C D = eq_quadrangle A B C D; E = on_pline E A C B, on_tline E B D A; F = midpoint F C B
A B C D E = pentagon A B C D E; F = on_pline F C D E, on_tline F A E D; G = lc_tangent G C B
A B C D = trapezoid A B C D; E = angle_mirror E B D C, eqdistance E D D C; F = shift F C D A
A B C D = eqdia_quadrangle A B C D; E = on_tline E A A D, on_tline E B D C; F = shift F E A D
A B C D = trapezoid A B C D; E = eqdistance E B D B, on_tline E B B D; F G = trisegment F G E A
A B C D = trapezoid A B C D; E = angle_mirror E C D A, eqdistance E A B C; F G = trisect F G B C E
A B C D E = pentagon A B C D E; F = angle_bisector F B D C, on_dia F B C; G = eqangle3 G A E B A C
A B C D = eq_trapezoid A B C D; E = angle_bisector E D C B, eqdistance E A D A; F = reflect F D A B
A B C D = eqdia_quadrangle A B C D; E = eqdistance E B C B, on_tline E D D C; F = eq_triangle F C A
A B C D = eq_quadrangle A B C D; E = angle_bisector E B D C, on_line E C B; F G = trisegment F G A E
A B C D = eq_quadrangle A B C D; E = angle_bisector E B C A, eqdistance E A C D; F = reflect F B C A
A B C D = isquare A B C D; E = angle_bisector E C A D, angle_bisector E D A C; F = on_circum F A B D
A B C D = rectangle A B C D; E = angle_bisector E B C D, eqdistance E D D B; F G = trisect F G A B E
A B C D = r_trapezoid A B C D; E = angle_bisector E B D C, on_pline E D C A; F G = trisect F G C E D
A B C D = quadrangle A B C D; E = on_circum E C D B; F G H I = rectangle F G H I; J = midpoint J I B
A B C D = eqdia_quadrangle A B C D; E = eqdistance E B B A, eqdistance E D A C; F G = trisect F G D A E
A B = segment A B; C D E = r_triangle C D E; F G = trisect F G B D A; H = lc_tangent H E A, on_tline H F E D
A B C D = r_trapezoid A B C D; E = shift E B A C; F = on_pline F D B A, eqangle3 F E D A C E; G = eqdistance G F A D
A = free A; B C D E = trapezoid B C D E; F = foot F C E D; G = on_aline G D F C F E, on_tline G C A B; H = on_dia H A G
A B C D = trapezoid A B C D; E F G = triangle E F G; H = on_circum H E F D; I = eqdistance I A F B, on_tline I A F C; J = midpoint J G I
A B C = triangle A B C; D = angle_mirror D B A C, eqdistance D C B A; E F = trisect E F C A B; G = angle_mirror G E D F, eqdistance G E A C
A B C = ieq_triangle A B C; D = eqdistance D A C B, lc_tangent D B C; E F = trisect E F D C B; G H I J = eq_trapezoid G H I J; K = shift K C G I
A B C = triangle A B C; D = angle_bisector D B C A, angle_bisector D C B A; E F = trisect E F A C B; G = angle_bisector G D C A, eqangle3 G C E E C F
A B C D = eq_quadrangle A B C D; E = eqdistance E A B D, on_tline E D C D; F = excenter F C D B; G = on_line G C F, on_dia G A F; H = reflect H B C F
A B C D = eqdia_quadrangle A B C D; E = eqdistance E B C A, on_tline E B A B; F G = trisect F G C D B; H = lc_tangent H B C, eqangle3 H B D C E A; I = on_circum I G H D
```

Figure 7: SAMPLED GEOMETRY LOGICAL SHAPES FOR EUCLID TRAINING

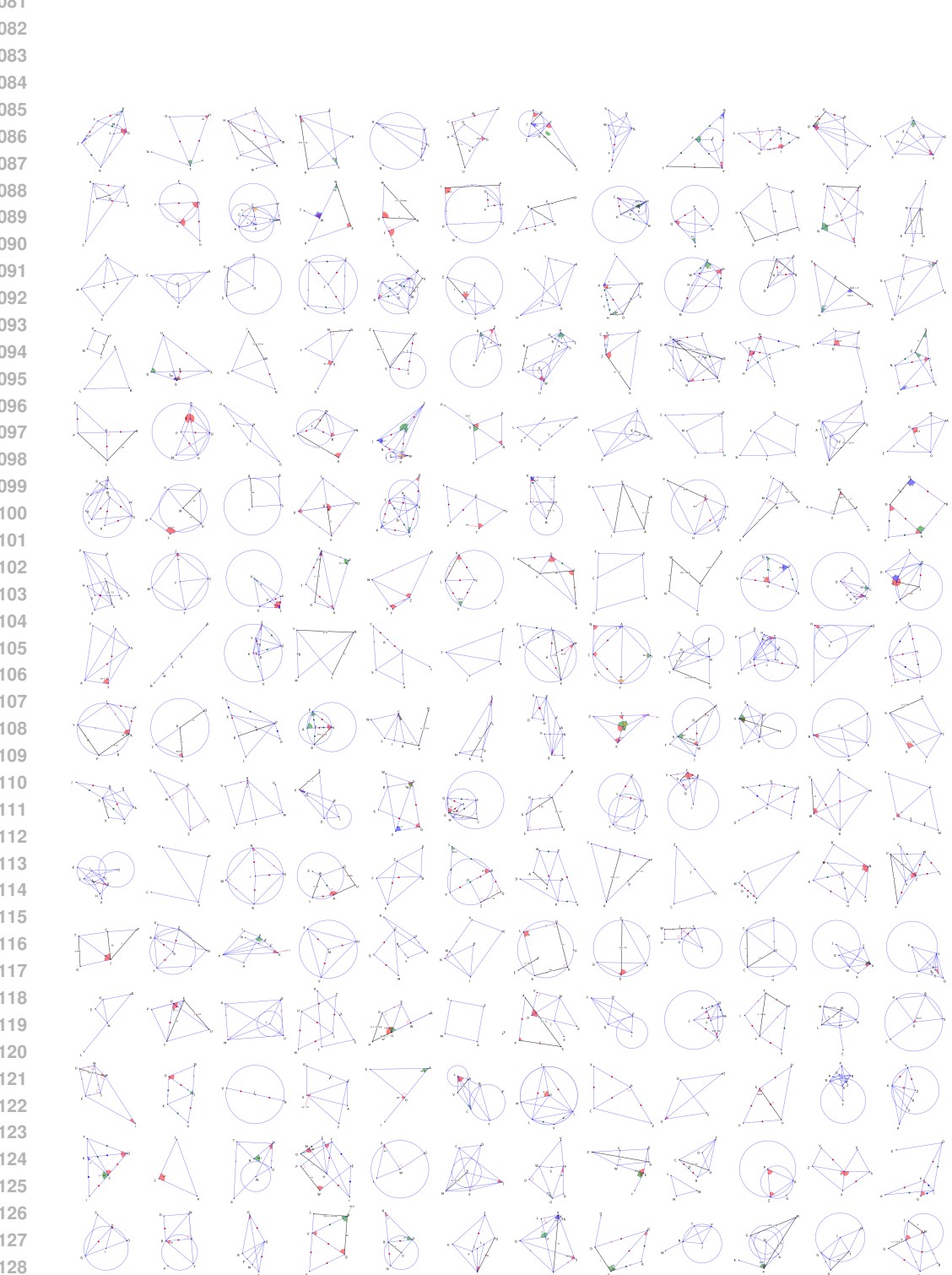

Figure 8: Examples of the geometry diagrams generated by our synthetic dataset engine.

## C.1 OPENAI-O3 MODEL EXAMPLE.

In Fig. 9, we include one example of OpenAI o3 (OpenAI, 2025) model's reasoning path in solving one problem from the Geoperception benchmark. We highlight the most obvious low-level visual perception error in their reasoning process, showing that low-level visual perception is still a challenge even for advanced 'thinking with image' models like o3.

**o3**

**Question: What is the point lying on line UR? other than UR themselves?**

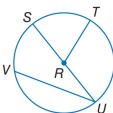

I'm looking at a circle with some interesting points: S at the top left, T at the top right, V in the middle left, and U at the bottom right. There's also point R somewhere inside, maybe at the center. I see three radii (SR, RT, RU), possibly the same length. I'm also noticing a chord from V to U and possibly another line extending inward from V, which might intersect the circle at the bottom. I'll need to zoom in to clarify the details better.

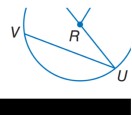

I'm looking at a circle with points S, T, V, and U placed around the perimeter. Point R is inside, possibly near the center. There are three radii connecting R to S, T, and U. I'm also noticing some interesting chords: one from V to some point on the circle's boundary, and another line from V that seems to hit point R before reaching inside. It's a little unclear, so I need to zoom in for more details and better understand these connections.

I'm looking at the line from R to U, and it seems to continue further, possibly showing a relationship with other parts of the diagram. I really need to inspect the entire setup more closely, making sure I don't miss any important connections or intersections. This could help me understand how everything is interconnected better before drawing any conclusions. Let's check the whole thing to see if I overlooked anything.

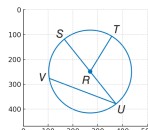

I see the line from R to U, and it seems to continue. There's also a chord from V that might intersect near the extended line from R to U. It looks like V's chord runs to the bottom-right, a bit above U. This would connect near where R to U intersects the circle. If the chord from V were drawn, it might pass inside the circle near R. Let me zoom in and check these relationships more clearly.

The visible line from V to the bottom-right does seem to pass through the intersection where line R→U meets the circle, which suggests V→U is indeed a chord. Now, I'm wondering about the line from R to V. There's no direct line drawn, but V to U might intersect something along R→U. I'm also considering whether lines like S→R intersect here. It looks like the line R→U only intersects at point U, but S→R might connect with it near R. The geometry is complex, but I'm getting closer to understanding it.

The question asks about the point on line UR excluding R and U. It seems they're referring to a point like T, but T doesn't appear to be on line UR. I've estimated coordinates for the points: R = (300, 230), U = (360, 380), V = (110, 270). Drawn lines like RS, RT, and RU do not seem to intersect at any additional points besides R and U. If referring to chord UR, the answer might still be no, as no other points seem to lie on it.

The question asks about a point on line UR, other than U and R. I think the line might refer to a chord or diameter that intersects the circle at another point, possibly V. But wait, the chord from V doesn't seem to be VU—it could be VR—though the line from V to the intersection of RU might be separate from RU. I'll double-check by zooming in on the bottom area of the diagram to look for that intersection more closely.

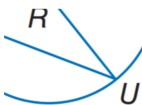

The question asks which point lies on line UR, other than U and R. This could mean the line is a chord or diameter passing through U and R, with another intersection point, maybe V. However, the line from V seems to enter the circle, potentially intersecting RU at a separate point. It seems like the line from V may be independent, not part of RU itself. I'll need to carefully examine the drawing again.

The line RU intersects with the chord VU at U, meaning both lines share this point. So, RU only contains R and U. The line extending through R and U may cut through the circle, reaching beyond U, potentially crossing the circle elsewhere, but that's not explicitly labeled. The question asks about "points" other than R and U, so the answer seems to be "no other labeled point" unless we consider other potential intersections like with line ST.

The user's question seems similar to one asked before, but with the clarification to exclude U and R. Initially, the answer might have been "R," but the user now specifies they don't want either endpoint. After calculating the slope of line UR, it appears that no other labeled points lie on the line, and the answer is most likely "none." The key here is understanding the user may want to clarify that only U and R lie on UR and that no other points are included—this could be explicitly stated.

**Answer: Within the diagram, no additional labeled point lies collinear with U and R; consequently, line UR contains only its two endpoints, U and R.**

Figure 9: An example of OpenAI o3 model's response to one Geoperception question.

## D  LIST OF VISUAL ENCODERS WE USE IN OUR EMPIRICAL STUDY.

The list of visual encoders that we use in our empirical study of the multimodal LLM design space is shown in Table 5 below.

Table 5: Summary of Visual Encoders

| Model | Params | Objective |
|---|---|---|
| ConvNeXt Large@512 | 200M | CLIP |
| ConvNeXt XXLarge@512 | 847M | CLIP |
| ViT-g/14@224 | 1.01B | CLIP |
| ViT-H/14@224 | 632M | CLIP |
| ViT-L/14@224 | 303M | CLIP |
| SigLIP@224 (ViT) | 428M | CLIP-like |
| DINOv2 Giant@224 (ViT) | 1.14B | Self-Sup |
| DINOv2 Large@224 (ViT) | 304M | Self-Sup |

# E   LLM USAGE STATEMENT

We have used LLMs for:

- Polishing writing at the sentence level.
- Establishing random baseline performance (with GPT-4o mini) and evaluation in Table 1 (with GPT-4o mini, GPT-4o, GPT-4.1, GPT-5-mini, GPT-5, Claude 3.5 Sonnet, Claude 4 Sonnet, Gemini-1.5-Flash, Gemini-1.5-Pro, Gemini-2.5-Flash).
- Synthesizing question template for training dataset generation (with Gemini-2.5-Pro).

## F    ADDITIONAL EXPERIMENTS

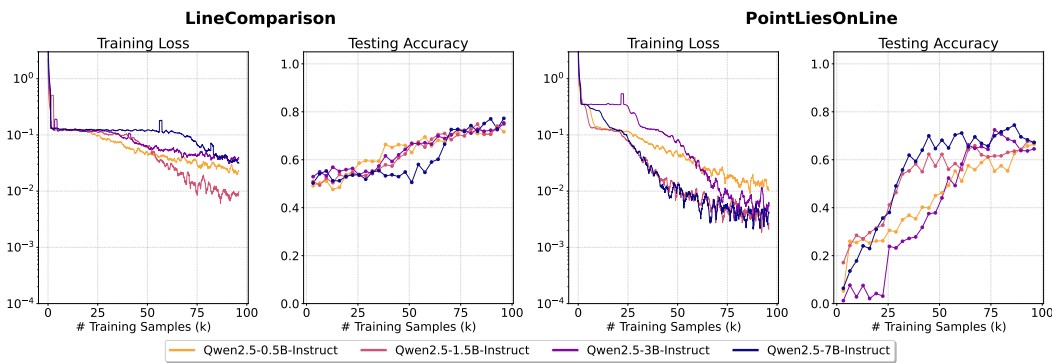

Figure 10: LLM size experiments. We show training loss and test accuracy (on corresponding Geoperception tasks), comparing four choices of LLM size with a fixed visual encoder and multimodal connector. Training losses are window-smoothed using a window size of 10 for better visibility.

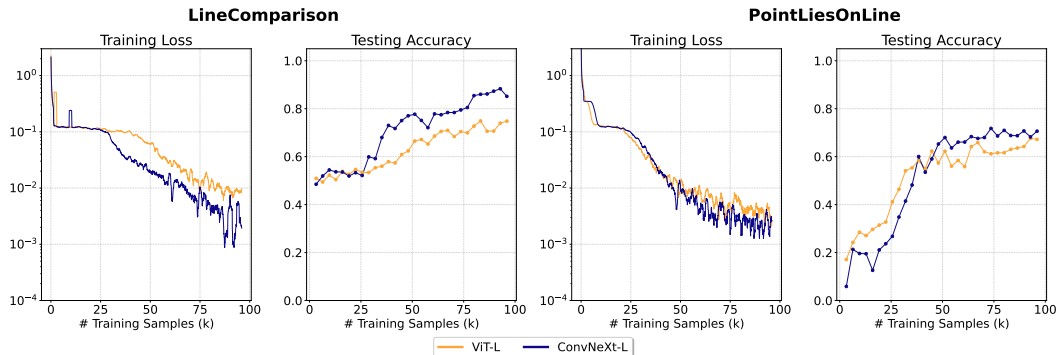

Figure 11: Vision encoder experiments. We show training loss and test accuracy (on corresponding Geoperception tasks), comparing two visual encoders, with a fixed multimodal connector and LLM. For a fair comparison, all visual encoder transcribe an image into 256 visual tokens. Training losses are window-smoothed using a window size of 10 for better visibility.

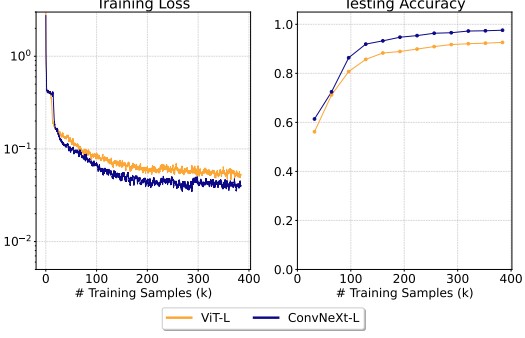

Figure 12: Vision encoder experiments conducted on all seven tasks with 43 unique geometry arrangements. We show training loss and test accuracy (on a in-domain holdout set), comparing two visual encoders, with a fixed multimodal connector and LLM. For a fair comparison, all visual encoder transcribe an image into 256 visual tokens. Training losses are window-smoothed using a window size of 10 for better visibility.

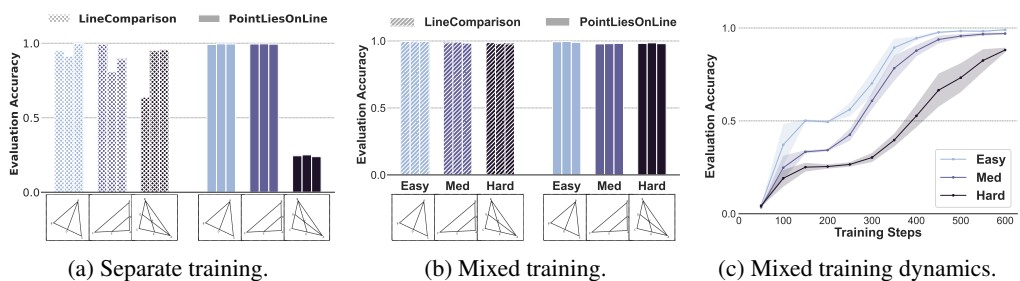

(a) Separate training.  (b) Mixed training.  (c) Mixed training dynamics.

Figure 13: Illustration of training on different data compositions. **(a)** We train the model on 96k examples separately on each problem difficulty level and report the evaluation accuracy on in-domain test points. **(b)** We train the model on 96k examples on the mixture of three problem difficulty levels and report the accuracy on each level separately. **(c)** In *PointLiesOnLine* task training on the mixture of three levels, we separately report the evaluation accuracy on each level, with respect to the number of training steps. In these plots, each bar represents one of three runs in an experiment.

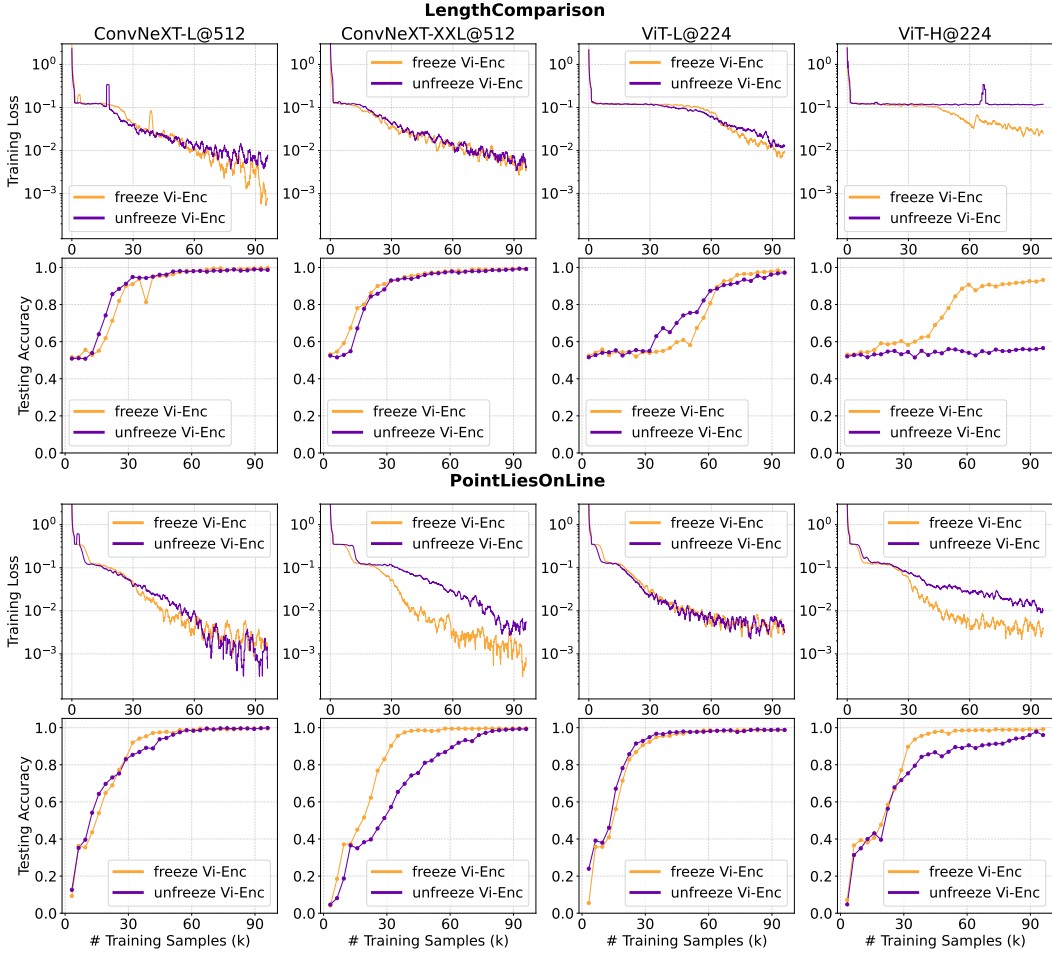

Figure 14: Tuning/freezing vision encoder experiments. We show training loss and test accuracy (on a in-domain holdout set), comparing freezing versus tuning the visual encoder during training across four visual encoders.

