# OpenReview forum: "Euclid: Lessons for Geometric Low-level Visual Perception in Multimodal LLMs"
_ICLR.cc/2026/Conference — Submitted to ICLR 2026_

### Official Review · Reviewer_AqGS · 2025-10-24

**Soundness:** 3
**Presentation:** 3
**Contribution:** 4
**Rating:** 8
**Confidence:** 4

**Summary:**

This paper analyzes the capability of MLLMs to identify and describe geometric relationships among geometric shapes in images. The authors introduce a new benchmark, Geoperception, and conduct experiments to investigate the low-level visual perception capability of MLLMs. The paper provides detailed analyses and identifies the benefits of curriculum learning and CNN-based visual encoders. Finally, the authors show that MLLMs trained on their dataset, Euclid-200k, achieve improved performance on out-of-distribution multimodal benchmarks.

**Strengths:**

* This paper focuses on the low-level visual perception of MLLMs, which is an important research topic for improving their performance and reliability.

* The analysis in Section 4 is well controlled and provides valuable insights for the future development of MLLMs. This paper presents useful suggestions for enhancing the low-level geometric perception of MLLMs, including the effectiveness of curriculum learning and CNN-based visual encoders. It also demonstrates that using larger LLMs or fine-tuning visual encoders has only limited influence.

**Weaknesses:**

I consider that this paper presents very useful observations, as mentioned in the Strengths section, but there are some concerns regarding the details of the dataset and experiments.

* **Rationale behind the task selection**. Among aspects of low-level geometric perception, this paper particularly focuses on the capability of MLLMs to accurately identify and describe geometric relationships within images. However, there is no clear explanation of why the paper focuses only on this aspect, rather than on other capabilities such as understanding individual geometric shapes.

* **Diversity of images in the dataset**. The introduced dataset is derived from Geometry-3K, which includes geometric shapes in a uniform format with the same color, line thickness, and text font and size. This level of uniformity makes it less suitable as an evaluation benchmark. Even within geometric shapes, there can be some variation in style and format, and it would be better for an evaluation dataset to include more diversity.

* **Missing citation**. Kamoi et al. (COLM 2025) [1] introduce a similar dataset. Their dataset is also designed to evaluate low-level geometric perception and includes tasks similar to AngleClassification and LineComparison. This paper should compare Geoperception with their dataset.

[1] Kamoi et al. (COLM 2025). VisOnlyQA: Large Vision Language Models Still Struggle with Visual Perception of Geometric Information. COLM 2025. https://openreview.net/forum?id=PYHwlyu2fa

---

The following additional experiments would strengthen the paper, but I do not prefer to request too many additional experiments during the rebuttal phase. The authors may prioritize experiments requested by other reviewers. If feasible, I would particularly like to see additional results for "Training: Model diversity.

* **Evaluation: Model diversity**. [1] reported that Gemini-2.5-Pro shows a substantial improvement in geometric perception compared to previous models. The paper would be strengthened if the authors also included results for this model. If there are constraints in budget, it would also be helpful to provide results on some challenging tasks such as POL.

* **Training: Model diversity**. I consider the analysis in Sections 4 and 5.2 to be a major contribution of this paper. However, the paper only evaluates models with the Qwen-2.5 family, which is not sufficient to support their claim. Evaluation with at least one more model family would largely strengthen the paper.

* **Training: Model scale**. As mentioned above, the experiment in Section 5.2 is conducted only with Qwen-2.5-VL-3B, which is a relatively small model. Experiments on larger models, even at the 8B scale, would strengthen the claim. *I understand the constraints in computational resources, so this point did not affect my final score.*

**Questions:**

* I would appreciate responses to the points I listed in the Weaknesses section.

---

> ### Author Response · Authors · 2025-11-21
>
> We thank the reviewer for the very positive assessment, and we are pleased that the reviewer finds our research question focused and important, our findings useful. We appreciate your valuable feedback and will try to address your concerns below:
>
>
> ### **Rationale behind the task selection**
>
> The rationale of selecting the tasks in Geoperception is derived from Euclid’s five axioms of plane geometry (please see our updated version of the paper). And all higher levels of geometric attributes and relationships can be derived from the basic axioms. For example, knowing the relationship between can help us determine the type of a geometry shape.
>
>
> ### **Diversity of images in the dataset**
>
> Thanks for pointing out the diversity of our dataset. Actually the images in Geometry-3K are not all in a uniformed format and the same color, for example, image1 contains green filling and red dashed line, image2 contains yellow background while some other images have no filling, blue lines and white background. There are also many combinations of different annotation types for angles and segments.
>
> Image1:
> https://huggingface.co/datasets/hiyouga/geometry3k/viewer/default/train?image-viewer=1E1571781D8C3D28910BEBF9938C4C1B632F5CD1
> Image2: https://huggingface.co/datasets/hiyouga/geometry3k/viewer/default/train?p=1&views%5B%5D=train&image-viewer=430394B1ACF3086BD67E6BC0CFA8D8C637513C7B
>
>
> ### **Missing citation**
>
> Thank you for bringing this paper to our attention. We have carefully read this paper and agree that Kamoi et al. similarly studied AngleClassification and LineComparison tasks and discussed a similar problem. We have cited the paper in our updated manuscript and discussed it as follows:
>
> _We show that careful selections of training curriculum and vision encoder can result in substantial performance gains, without scaling LLM backbone. Our findings are beneficial for practitioners with a low compute budget, or requiring deployment in edge devices with small models.
>
>
> ### **Evaluation on Gemini-2.5-pro**
>
> Thank you for suggesting these additional experiments (and flexibility :-). At the time of submission, we were not able to run Gemini 2.5 Pro experiments due to API rate limits. We now report the complete evaluation results for Gemini 2.5 Pro below:
>
> | Model                   | POL      | POC      | ALC      | LHC        | PEP      | PRA      | EQL      | Average   |
> |-------------------------|----------|----------|----------|------------|----------|----------|----------|-----------|
> | Random Baseline         | 1.35     | 2.63     | 59.92    | 51.36      | 0.23     | 0.00     | 0.02     | 16.50     |
> | Human                   | 100.00   | 98.00    | 98.00    | 96.00      | 98.00    | 100.00   | 90.00    | 97.14     |
> | Qwen-2.5-VL-3B          | 15.36    | 18.94    | 23.33    | 16.64      | 11.32    | 50.89    | 51.72    | 26.89     |
> | Qwen-2.5-VL-72B         | 50.39    | 72.42    | 56.27    | 72.24      | 33.36    | 68.87    | 60.32    | 59.13     |
> | GPT-5                   | 40.72    | 81.06    | 55.49    | 79.77      | 33.20    | 76.42    | 63.98    | 61.52     |
> | Claude 4 Sonnet         | 55.18    | 81.34    | 67.31    | 79.99      | 53.55    | 87.74    | 73.92    | 71.29     |
> | Gemini-2.5-Flash        | 44.56    | 84.12    | 65.62    | 80.92      | 62.85    | 93.40    | 65.67    | 71.02     |
> | Gemini-2.5-Pro          | 68.17| **86.35**| 78.84| 84.58      | **77.13**| **97.17**| **75.92**| **81.17** |
> | Euclid                  | **79.43**| 74.37    | **92.43**| **98.13**  | 68.06    | 76.42    | 62.08    | 78.70     |
>
> Consistent with observations in prior work, Gemini-2.5-Pro achieves a remarkable improvement in geometric perception over earlier MLLMs, though it still remains approximately 16% below human performance. Furthermore, our Euclid model trails Gemini-2.5-Pro by only 2.33%, despite being trained on substantially less data and having far fewer parameters. We have incorporated this into the updated version of the paper.
>
>
> ### **Model diversity and scale**
>
> Thank you for the valuable suggestions! We totally agree that expanding the study to larger model scales and additional model families would further reinforce the empirical findings and enhance the overall contribution. However, due to the limited time available during the rebuttal phase, we are unable to include these results at this moment. We plan to incorporate these extended experiments in a future version of the work.

---

> > ### Comment · Reviewer_AqGS · 2025-11-22
> >
> > Thank you for your response. While I still have concerns regarding the diversity of the images in the dataset and the comprehensiveness of the experiments, I maintain my positive assessment of this paper because I believe the analysis offers valuable insights for future research on the visual perception of LVLMs.

---

### Official Review · Reviewer_sdwW · 2025-10-28

**Soundness:** 3
**Presentation:** 3
**Contribution:** 2
**Rating:** 4
**Confidence:** 4

**Summary:**

The paper targets a well-know gap in MLLMs: low level geometric visual perception. The authors introduce Geoperception, a 2D geometry benchmark derived from real textbooks with seven task types, and show that leading open/closed models underperform humans. They also build a controlled synthetic geometry data engine to study training/architectural choices and report several lessons. Finally, they train Euclid (Qwen-2.5-VL-3B) on a 200k synthetic instruction dataset and report competitive gains on LLVP without obvious degradation on general tasks.

**Strengths:**

1. By isolating LLVP in 2D geometry and using a synthetic engine, the study can attribute effects to data/architecture rather than confounds from high-level semantics.
2. Geoperception spans seven tasks and evaluates fifteen major MLLMs with a transparent scoring rule, plus a human baseline.
3. Curriculum helps; CNN encoders excel at LLVP; and tuning the vision encoder is often unnecessary—practical guidance for building LLVP-sensitive MLLMs.

**Weaknesses:**

1. It’s almost a community consensus that multimodal models underperform on a specific task mainly because they lack task-specific data; this paper’s core contribution is essentially to reiterate that point. In my view, the novelty is insufficient.
2. As for the other findings—e.g., that CNN-based visual encoders are more suitable than ViT—this is not surprising. I suspect the result is largely due to insufficient data; with proper data scaling, the conclusion may not hold.
3. That MLLMs struggle with LLVP is already broadly observed; the novelty mainly lies in a geometry-focused, cleaner testbed and a systematized study, rather than a new principle or paradigm.

**Questions:**

How do your conclusions change under explicit domain randomization on the rendering/annotation pipeline?

---

> ### Author Response · Authors · 2025-11-21
>
> Thank you for reviewing our paper. We are pleased to see that you find our study focused and our evaluation comprehensive. We appreciate your valuable feedback, and we will try to address your concerns below:
>
> ### **Is this paper just to improve performance by adding task-specific data?**
>
> We would like to clarify that ‘just adding specific dataset to improve task-specific performance’ is not our main contribution. In fact, our initial experiment shows the opposite: naively adding task-specific data does not yield meaningful improvements (Figure 2a). In addition, as we mentioned in line 183, Cambrian-1 (and probably many other SoTA MLLMs), which has already been trained on Geo-170K, a dataset that contains exactly the same image set as Geoperception, still doesn’t outperform other MLLMs. In contrast, Euclid achieves performance exceeding even the strongest proprietary models on Geoperception, further indicating that the gains cannot be attributed to mere dataset inclusion.
>
> ### **Whether CNN-based architecture outperforms ViT is surprising and useful and the sufficiency of our training dataset**
>
> To our knowledge, we are the first study showing CNN-based visual encoders perform on-par or even better than ViT on low level visual perception tasks for MLLMs, and this is achieved by our fully controlled experiment (identical LLM, number of visual tokens and dataset) on a task that all models initially struggle with (demonstrated in Figure 4’s testing accuracy). We kindly argue that the dataset amount we use for this study (96K) is considered sufficient due to the final convergence of the best models >90% percent final accuracy. We think the current experimental setting is important for practical applications, since these tasks represent fundamental aspects of geometric understanding on which current models still struggle. Furthermore, as demonstrated in Lesson 1, the model fails to learn such tasks even when provided with large amounts of in-domain training data.
>
> To further address your concern, we repeat the experiment comparing ViT and ConvNeXt under the joint-training setting across all seven tasks with 43 unique geometry arrangements, and report both the training loss and the in-domain testing accuracy.  The results, which are presented in Figure 12 of the updated version of the paper, show that ConvNeXt converges more quickly and achieves a higher final testing accuracy than ViT, which is consistent with the conclusion drawn from the main experiments in the paper.
>
> ### **Main contribution of this paper**
>
> We agree that the main contribution and novelty of our work lies in the clean testbed and conclusions gained by the systematized study, rather than introducing a new principle or paradigm. We think this is important because although this is broadly observed as a problem, there is only limited insight into why the sota models still struggle with it. For example, how to design the dataset to optimize the performance on a new domain (Lesson 1), whether we should scale up LLM size, change visual encoders and unfreeze them to optimize the performance on LLVP (Lessons 2,3,4). We believe the lessons we learn from our controlled study are not broadly known, and provide useful information for model developers to design their dataset composition and model architectures.

---

### Official Review · Reviewer_1TVt · 2025-11-01

**Soundness:** 1
**Presentation:** 3
**Contribution:** 2
**Rating:** 2
**Confidence:** 5

**Summary:**

This paper introduces Geoperception, a benchmark revealing that current multimodal large language models struggle with geometric low-level visual perception, such as identifying collinearity, angle types, or line lengths. To address this, the authors create a synthetic 2D geometry dataset engine and conduct controlled experiments, uncovering four key lessons. Additionally, they train Euclid, a 3B-parameter model on a 200k synthetic dataset, resulting in improved performance on Geoperception while retaining general capabilities.

**Strengths:**

1. Well-motivated problem: The paper clearly articulates a real and consequential gap in MLLM capabilities: LLVP, which impacts applications like mathematical reasoning, robotics, and medical imaging.
2. Actionable insights: The four lessons provide concrete guidance for improving LLVP in MLLMs, some of which challenge previous assumptions.

**Weaknesses:**

The major problem of this paper is its poor soundness. As claimed by the authors, contribution of this work is threefold: the proposal of Geoperception, the conclusion of four lessons for LLVP, and training the Euclid model. However, none of these contributions are sufficiently solid.
1. In the Geoperception benchmark, the proposed task types may not be able to cover geometric perception capabilities. For example, the authors mention task types PointLiesOnLine and PointLiesOnCircle, which are designed for (non-numeric) geometric shape understanding. However, there may be many other task types belonging to this category, including but not limited to: determining the type of a specific shape (triangle, rectangle, circle, etc.), identifying the geometric relationship between two shapes (parallel, similar, tangent, etc.), or estimating the size and position of geometric shapes.
2. The four lessons are rather arbitrary with irresponsible overclaim. First of all, all ablation studies only include two subtasks. It is extreme overclaim that such narrow taskset could represent LLVP. Besides, each lesson seems to be established in unreliable experiment settings:
  - Lesson 1: the dataset sizes of different settings are not equal. While for each difficulty level the dataset size used for training is 32k, the mixed configuration uses 96k training data in total. Therefore, it is unclear whether performance gains should be contributed to the inclusion of samples from different difficulty levels or simply the larger dataset size.
  - Lesson 2: Most models converge to nearly 100% accuracy on the two subtasks, and it is insufficient to judge the capability of models solely by their convergence speed.
  - Lesson 3: Similarly, ViT-g and ViT-L share extremely close terminating performance with convolution-based visual encoders.
  - Lesson 4: The experiment only includes convolution-based visual encoders, while the popular ViT-based ones are uncovered.
3. In sec. 5, the cliam "endow multiple geometric low-level visual perception capabilities into a generalist MLLM" is also unestablished.
  - The improvements in Table 3 are trivial. Geoperception improvements are trivial because Geoperception and Euclid share identical (or at least very similar) data distributions. It is expected that training on in-domain data improves test performance. For the general benchmarks, compared with Qwen-General-Only, Euclid* exhibits only neglegible improvements (<1%). Fluctuations brought by different training seeds might even achieve greater difference.
  - The scope of the incorporated benchmark is limited. It is helpful to add datasets such as GeoQA, Math-Vista or We-Math.

**Questions:**

1. In line 275-276, what's the meaning of "a mixture of **three** geometric shapes"? Does it mean that there are only three fixed geometric arrangements for all the images, along with Figure 2? If so, the overclaim problem for the lessons will be even more severe since the data diversity is also limited in addition to limited task type; if not, what are the specific criteria for splitting easy, middle and hard sets in Sec. 4.1?
2. Why not include Gemini-2.5-pro in Table 1 while the authors have already used it in dataset construction in Sec. 5.1? As the generally strongest MLLM, it will be helpful to report its performance on geometric perception.

---

> ### Author Response · Authors · 2025-11-21
>
> Thank you for reviewing our paper. We are pleased that you find our research question well-motivated and our insights actionable. We will try to address your concerns below:
>
>
> ### **Geoperception tasks**
>
> Thanks for raising this point. First of all, geoperception does include the LengthComparision for segment size estimation and Parallel/Perpendicular recognition questions. Second, we acknowledge that additional task types—such identifying similar triangles—also fall within the broader scope of LLVP. Nevertheless, our objective is not to exhaustively enumerate all tasks in this category, which. The rationale of selecting the tasks in Geoperception is derived from Euclid’s five axioms of plane geometry (please see our updated version of the paper in Section 3). And all higher levels of geometric attributes and relationships can be derived from the basic axioms. For example, knowing the relationship between can help us determine the type of a geometry shape. In addition, introducing more tasks would diminish readability without offering additional analytical benefit.
>
>
> ### **Dataset amount for Lesson 1**
>
> The training dataset size for each difficulty level is controlled to 32K for both separate and mixed training. We believe these two settings are comparable because they use the same amount and distribution of training data to (1) separately train three models, and (2) jointly train one model, both evaluated on the same in-domain dataset across the three difficulty levels.
>
> To further address your concern, we increase the dataset size to 96K for separate training and include the new result in Figure 13 of the updated version of the paper. Note that in this setting, the training dataset amount for each difficulty level is now three times larger than in mixed training. Regardless, **we still observe that the model struggles to learn the hard tasks**, especially on PointsOnLine. This result further demonstrates that incorporating simpler shapes aids in learning more difficult data and enables the model to achieve near-perfect accuracy on tasks that it fails to learn from scratch even with three times larger in-domain dataset.
>
>
> ### **Near perfect final performance on Lesson 2 and 3**
>
> Thanks for raising this point. We agree that in Lessons 2 and 3, most model variants eventually converge to near-perfect performance on the two tasks. However, our argument here focuses on training efficiency without scaling LLM backbone. We believe our findings are beneficial for practitioners with a low compute budget, less data, or requiring deployment in edge devices with small models.
>
> Nonetheless, we acknowledge that incorporating tasks on which the model does not achieve perfect terminating performance could further strengthen the empirical findings. **Therefore, we adopt Geoperception as the out-of-domain evaluation throughout training and repeat our experiments on LLM size and visual encoder.** We have included these results in the Figure 10 and 11 of the updated version of the paper. From the new experiment, we find that: 1. Scaling LLM size doesn’t have remarkable benefits on out-of-domain generalization and 2. CNN-based visual encoder attains faster convergence on the LineComparison task and demonstrates overall better out-of-domain performance across the two evaluated tasks. These observations remain consistent with the conclusions drawn from the main experiment. Notably, although most visual encoders achieve near-perfect in-domain performance in Figure 4, we find that a lower final training loss is associated with better out-of-domain generalization, as demonstrated in Figure 11.
>
> Lastly, the fact that the model reaches at most 80% terminating performance on Geoperception, even with a large amount of training data, further illustrates the discrepancy between the dataset distributions produced by our dataset generation engine and those underlying Geoperception.

---

> ### Author Response · Authors · 2025-11-21
>
> ### **Including ViT visual encoders in Lesson 4**
>
> Thank you for the suggestion. We have incorporated the corresponding results into Figure 14 of the revised paper. The figure reports both the training loss and the testing accuracy for four visual encoders, comprising two CNN-based and two ViT-based ones. The empirical evidence indicates that varying the choice of visual encoder does not yield a substantial advantage in solving LLVP tasks, which is consistent with the conclusion drawn in the main paper.
>
>
> ### **Performance improvement in Table 3**
>
> Our goal for Table 3 is to demonstrate that MLLM’s performance on some LLVP tasks can be significantly improved **without compromising** their general multimodal capabilities with proper data composition, rather than improving the model’s general capability. In addition, as is emphasised in the footnote of page 8, the questions and images from Geoperception are fundamentally different and much more diverse than our synthetic dataset, since they are derived from real textbook diagrams, has many shapes and manual annotations that are not reproducible by our dataset engine.
>
> In addition, following your suggestion, we have expanded our evaluation to include GeoQA, Math-Vista, and We-Math below. Across these benchmarks, the Euclid-Star model shows notable improvements over its baseline, Qwen-2.5-VL-3B, achieving gains of 10.18% and 6.69%, respectively, on We-Math and GeoQA. We believe these results indicate that, with deliberate dataset composition, large-scale synthetic data not only significantly enhances MLLMs’ low-level visual perception without compromising their general multimodal capabilities, but also provides **notable benefits for geometry-math reasoning, even in the absence of task-specific visual reasoning training**. We have included the result in Table3 and discussed in Section 5 in the updated version of the paper.
>
>
> | Model              | MathVista | We-Math | GeoQA |
> |--------------------|-----------|---------|-------|
> | Random Baseline    | 17.90     | 21.59   | 25.00 |
> | Qwen-2.5-VL-3B      | **52.08** | 35.34   | 51.48 |
> | Qwen-General-Only   | 51.66     | 38.33 | 53.43 |
> | Euclid             | 50.23     | 31.90   | 38.26 |
> | Euclid-Star           | 51.38     | **42.13** | **61.66** |
>
>
>
> ### **Shapes and Tasks used for empirical study**
>
> In our empirical study, our training and evaluation dataset are both generated by three geometry logical shapes. For each specific geometry logical shape, our dataset generation engine generates large amounts of geometry shapes that are different from each other, despite sharing the same logical arrangements. To further address your concern, we repeated a key experiment comparing ViT and ConvNeXt under the joint-training setting across all seven tasks with 43 unique geometry arrangements, and reported both the training loss and the in-domain testing accuracy.  The results, which are presented in Figure 12 of the updated version of the paper, show that ConvNeXt converges more quickly and achieves a higher final testing accuracy than ViT, which is consistent with the conclusion drawn from the main experiments in the paper.

---

> > ### Author Response · Authors · 2025-11-21
> >
> > ### **Evaluation on Gemini-2.5-pro**
> >
> > We fully agree that incorporating the most recent state-of-the-art MLLMs into our Geoperception benchmark strengthens the evaluation. At the time of the original submission, we were unable to run Gemini-2.5-Pro due to API rate limits. We have now completed these experiments and report the full set of results below:
> >
> >
> > | Model                   | POL      | POC      | ALC      | LHC        | PEP      | PRA      | EQL      | Average   |
> > |-------------------------|----------|----------|----------|------------|----------|----------|----------|-----------|
> > | Random Baseline         | 1.35     | 2.63     | 59.92    | 51.36      | 0.23     | 0.00     | 0.02     | 16.50     |
> > | Human                   | 100.00   | 98.00    | 98.00    | 96.00      | 98.00    | 100.00   | 90.00    | 97.14     |
> > | Qwen-2.5-VL-3B          | 15.36    | 18.94    | 23.33    | 16.64      | 11.32    | 50.89    | 51.72    | 26.89     |
> > | Qwen-2.5-VL-72B         | 50.39    | 72.42    | 56.27    | 72.24      | 33.36    | 68.87    | 60.32    | 59.13     |
> > | GPT-5                   | 40.72    | 81.06    | 55.49    | 79.77      | 33.20    | 76.42    | 63.98    | 61.52     |
> > | Claude 4 Sonnet         | 55.18    | 81.34    | 67.31    | 79.99      | 53.55    | 87.74    | 73.92    | 71.29     |
> > | Gemini-2.5-Flash        | 44.56    | 84.12    | 65.62    | 80.92      | 62.85    | 93.40    | 65.67    | 71.02     |
> > | Gemini-2.5-Pro          | 68.17| **86.35**| 78.84| 84.58      | **77.13**| **97.17**| **75.92**| **81.17** |
> > | Euclid                  | **79.43**| 74.37    | **92.43**| **98.13**  | 68.06    | 76.42    | 62.08    | 78.70     |
> >
> > Consistent with observations in prior work, Gemini-2.5-Pro achieves a remarkable improvement in geometric perception over earlier MLLMs, though it still remains approximately 16% below human performance. Furthermore, our Euclid model trails Gemini-2.5-Pro by only 2.33%, despite being trained on substantially less data and having far fewer parameters. We have incorporated this into the updated version of the paper.

---

### Official Review · Reviewer_pYoq · 2025-11-01

**Soundness:** 4
**Presentation:** 3
**Contribution:** 3
**Rating:** 6
**Confidence:** 4

**Summary:**

In this paper, the authors explore the issue of low-level visual perception (LLVP) of VLMs. They start by building a dataset with QA pairs for geometry understanding and demonstrate the significant shortcomings of VLMs. Then, they try to propose some basic ideas to test how to improve the model on these datasets, including Lesson 1: Exposure to simpler visual perception data aids in learning difficult LLVP. Lesson 2: Scaling LLM size yields limited gains in LLVP learning. Lesson 3: CNN-based visual encoders enable more efficient learning of LLVP tasks. Lesson 4: Tuning the vision encoder does not offer a strong advantage in learning LLVP. The authors further extend the dataset and train a model; results show that the proposed model achieves high performance on the LLVP dataset and can improve the performance on real-world datasets as well.

**Strengths:**

1. LLVP is a very fundamental problem of VLMs, and without this capability, the VLM's performance on complex tasks can not be guaranteed.
2. The paper is very well written and easy to follow.
3. The lessons are useful for the future development of the VLMs.
4. The proposed model is effective and can beat the performance of strong models on math reasoning datasets.

**Weaknesses:**

1. Dataset validation. Geoperception is mostly synthetic, so the performance of the dataset largely reflects the performance on certain tasks and rules. It is not very convincing to say LLVP is bad for VLMs, given that the test results are based on a synthetic dataset. Thus, some human-annotated or drawn samples are helpful.

2. Dataset choices in experiments. All experiments use LC and PLOL, narrowing the generalizability of the lessons. It might be better to run them on all datasets or change to some other dataset combinations for other lessons.

3. Lacks related work and novelty. For instance, VisonlyQA [1] demonstrates a very similar conclusion and is not cited. Considering VisonlyQA, the novelty of this paper should be adjusted to a lower level.

[1] Kamoi, Ryo, et al. "Visonlyqa: Large vision language models still struggle with visual perception of geometric information." COLM 2025

**Questions:**

See above.

---

> ### Author Response · Authors · 2025-11-21
>
> Thank you for reviewing our paper. We appreciate your positive feedback, and we are pleased that you find our research question to address fundamental problems in VLMs, the paper to be well written and easy to follow, the lessons to be valuable, and the models to be effective. We will try to address your concerns below:
>
> ### **Dataset validation**
>
> We agree that dataset diversity and proximity to real-world conditions are essential for a reliable evaluation and this principle guided the design of Geoperception. **Rather than relying on synthetic data, all images in Geoperception originate from real-world, textbook-style geometry datasets that feature diverse geometric shapes, background colors, and annotation styles, including human-drawn annotations.** Moreover, all geometric relations are manually annotated and reorganized into large-scale question–answer pairs to support systematic and robust evaluation.
>
> ### **Task choices**
>
> Thanks for pointing this out. We chose PLOL and LC as our test bed because both tasks remain challenging for current MLLMs and represent two of the most fundamental geometric skills: basic spatial relationship reasoning and basic spatial measurement.
> To demonstrate the generalizability of our empirical analysis and address your concern, we repeat a key experiment comparing ViT and ConvNeXt but under the joint-training setting across all seven tasks with 43 unique geometry arrangements, reporting both the training loss and the testing accuracy. The results, presented in Figure 12 of the updated version of the paper, show that ConvNeXt converges faster and attains a higher final testing accuracy than ViT. These observations remain consistent with the conclusions drawn from our main experiments.
>
>
> ### **Related work**
>
> Thank you for bringing this paper to our attention. We agree that Kamoi et al. similarly studied AngleClassification and LineComparison tasks. We have cited the paper in our updated manuscript and discussed it as follows:
>
> _We show that careful selections of training curriculum and vision encoder can result in substantial performance and efficiency gains, without scaling LLM backbone. Our findings are beneficial for practitioners with a low compute budget, or requiring deployment in edge devices with small models._

---

### Author Response · Authors · 2025-11-21
**General Response**

We thank the reviewers for valuable and constructive comments. We will address shared concerns of the reviewers below.

1.  Concerns that the dataset is too synthetic (pYoq), lacks diversity (AqGS), or that the selected tasks are arbitrary/limited (1TVt, AqGS)

For Geoperception, rather than relying on synthetic data, all images in Geoperception originate from real-world, textbook-style geometry datasets that feature diverse geometric shapes, background colors, and annotation styles, including human-drawn annotations. Moreover, all geometric relations are manually annotated and reorganized into large-scale question–answer pairs to support systematic and robust evaluation.

For empirical study, following the reviewer’s suggestion, we adopt Geoperception as the out-of-domain evaluation throughout training and repeat our experiments on LLM size and visual encoder, the result is included in Figure 10 and Figure 11 in the updated version of the paper. Our observations remain consistent with the conclusions drawn from the main experiment. Notably, although most visual encoders achieve near-perfect in-domain performance in Figure 4, we find that a lower final training loss is associated with better OOD generalization for CNN-based visual encoders, as demonstrated in Figure 11.

In addition, we also repeat the experiment comparing ViT and ConvNeXt but under the joint-training setting across all seven tasks with 43 unique geometry arrangements, reporting both the training loss and the testing accuracy. The results, presented in Figure 12 of the updated version of the paper, show that ConvNeXt converges faster and attains a higher final testing accuracy than ViT. These observations also remain consistent with the conclusions drawn from our main experiments.


2. Limited novelty in scaling dataset size, and performance gains are restricted to in-domain evaluations.

Following the reviewers’ suggestions, we have additionally evaluated Euclid models on We-Math and GeoQA datasets, both of which are general mathematical reasoning benchmarks that are OOD. Euclid models demonstrate marked improvements when trained on a mixture of our dataset and general multimodal reasoning datasets. This observation suggests that low-level perception can be a bottleneck for mathematical reasoning, and our dataset directly addresses this challenge.

| Model               | Geoperception | MMBench | MMStar | MathVista | Math-Vision | We-Math | GeoQA |
|---------------------|---------------|---------|--------|-----------|-------------|---------|-------|
| Random Baseline     | 16.37         | 2.90    | 24.60  | 17.90     | 5.86        | 21.59   | 25.00 |
| Qwen-2.5-VL-3B       | 26.89         | 77.84   | 47.87  | **52.08** | 15.92       | 35.34   | 51.48 |
| Qwen-General-Only    | 30.00         | 81.01   | 47.73  | 51.66     | **16.79**   | 38.33   | 53.43 |
| Euclid               | **78.70**     | 74.87   | 43.20  | 50.23     | 13.87       | 31.90   | 38.26 |
| Euclid-Star             | 78.53         | **81.52** | **48.13** | 51.38 | **16.79** | **42.13** | **61.66** |

---

> ### Author Response · Authors · 2025-11-21
>
> 3. Missing Gemini-2.5-Pro Performance
>
> We fully agree that incorporating the most recent state-of-the-art MLLMs into our Geoperception benchmark strengthens the evaluation. At the time of the original submission, we were unable to run Gemini-2.5-Pro due to API rate limits. We have now completed these experiments and report the full set of results below:
>
>
> | Model                   | POL      | POC      | ALC      | LHC        | PEP      | PRA      | EQL      | Average   |
> |-------------------------|----------|----------|----------|------------|----------|----------|----------|-----------|
> | Random Baseline         | 1.35     | 2.63     | 59.92    | 51.36      | 0.23     | 0.00     | 0.02     | 16.50     |
> | Human                   | 100.00   | 98.00    | 98.00    | 96.00      | 98.00    | 100.00   | 90.00    | 97.14     |
> | Qwen-2.5-VL-3B          | 15.36    | 18.94    | 23.33    | 16.64      | 11.32    | 50.89    | 51.72    | 26.89     |
> | Qwen-2.5-VL-72B         | 50.39    | 72.42    | 56.27    | 72.24      | 33.36    | 68.87    | 60.32    | 59.13     |
> | GPT-5                   | 40.72    | 81.06    | 55.49    | 79.77      | 33.20    | 76.42    | 63.98    | 61.52     |
> | Claude 4 Sonnet         | 55.18    | 81.34    | 67.31    | 79.99      | 53.55    | 87.74    | 73.92    | 71.29     |
> | Gemini-2.5-Flash        | 44.56    | 84.12    | 65.62    | 80.92      | 62.85    | 93.40    | 65.67    | 71.02     |
> | Gemini-2.5-Pro          | 68.17| **86.35**| 78.84| 84.58      | **77.13**| **97.17**| **75.92**| **81.17** |
> | Euclid                  | **79.43**| 74.37    | **92.43**| **98.13**  | 68.06    | 76.42    | 62.08    | 78.70     |
>
>
> Consistent with observations in prior work, Gemini-2.5-Pro achieves a remarkable improvement in geometric perception over earlier MLLMs, though it still remains approximately 16% below human performance. Furthermore, our Euclid model trails Gemini-2.5-Pro by only 2.33%, despite being trained on substantially less data and having far fewer parameters. We have incorporated this into the updated version of the paper.
>
> 4. Lacked citation of Kamoi et al.
>
> We agree that Kamoi et al. similarly studied AngleClassification and LineComparison tasks. We have cited the paper in our updated manuscript (Section 2) and discussed it as follows:
>
> _We show that careful selections of training curriculum and vision encoder can result in substantial performance and efficiency gains, without scaling LLM backbone. Our findings are beneficial for practitioners with a low compute budget, or requiring deployment in edge devices with small models._

---

### Meta-Review · Area_Chair_9aBX · 2026-01-08

**Summary:**

This paper constructs a dataset to examine the low-level visual perception capabilities of VLMs and discusses several key factors affecting performance, including training with a data curriculum, scaling effects, the use of CNN-based encoders, and the impact of fine-tuning visual encoders. The authors further fine-tune models using the proposed dataset and demonstrate performance improvements.

I believe the major concern of this paper, which is shared by most reviewers, is whether its novelty is sufficient. As noted by many reviewers, the observation that VLMs perform poorly on low-level visual perception is already a common consensus in the research community. While the paper presents a systematic analysis, it does not seem to be comprehensive enough. One potential way to strengthen the paper would be to include a broader range of tasks (see Reviewer 1TVt’s comment) with analysis for a deeper understanding. Another way would be to provide principled methods for improving models beyond data-only solutions.

Based on this, I recommend a weak reject.

**Reviewer Concerns:**

- (Addressed) Some reviewers raised concerns about whether the constructed data are sufficiently representative, given that they are synthesized. The authors provided arguments addressing this issue, which seem acceptable to me.
- (Addressed) Reviewers were concerned that the experimental setup did not fully support the claims. The authors provided additional results following the reviewers’ suggestions.
- (Not addressed) Novelty concerns, as shortcomings in low-level visual perception are already a common consensus. The authors may want to provide a more comprehensive study covering broader tasks and deeper analysis, or propose more principled methods for improvement.
- (Addressed) Other clarification questions were properly addressed.

**Reviewer Scores:**

I believe all reviewers will keep their scores.

---

### Decision · Program_Chairs · 2026-01-26

Reject